# TUG protein acts through a disordered region to organize the early secretory pathway

Anup Parchure [1,2] ✉, Helen Tejada [1], Zhiqun Xi[2], Yeongho Kim [2], Maohan Su [2], You Yan [], Omar Julca-Zevallos[1,2,5], Abel R. Alcázar-Román[1,6], Marie Villemeur[3], Xinran Liu[2], Derek Toomre [2], Ishier Raote [3] & Jonathan S. Bogan [1,2,4] ✉

The Endoplasmic Reticulum (ER)-Golgi Intermediate Compartment (ERGIC) is a network of tubules and vesicles known for producing COPI vesicles and receiving COPII vesicles from the ER. Much about its identity, stability, and regulation remains unknown. Here, we show that TUG (UBXN9, Aspscr1) protein, a central regulator of GLUT4 trafficking, localizes to the ERGIC, and that its deletion enhances anterograde flux of a model soluble cargo protein. TUG deletion redistributes ERGIC markers to the cis-Golgi and alters Golgi morphology. TUG forms biomolecular condensates in vitro and contains a central disordered region that mediates its recruitment to ERGIC membranes. A distinct N-terminal region mediates its oligomerization in cells. TUG deletion disrupts ERGIC-dependent processes, including autophagy and collagen secretion, and alters the targeting of the CFTR chloride channel. We conclude that TUG organizes and stabilizes ERGIC membranes to support their roles in diverse secretory and degradative membrane trafficking pathways.

About a quarter of the proteome enters the secretory pathway in mammalian cells. These proteins, including both soluble and membrane-associated secretory proteins, are synthesized at the endoplasmic reticulum (ER). Properly folded proteins are packaged into COPII-coated vesicles at ER exit sites (ERES)[1]. After uncoating, these vesicles coalesce with each other to form the ER-Golgi intermediate compartment (ERGIC) and Golgi complex. The ERGIC may be a transient intermediate compartment that arises as a result of rapid flux between the ER and Golgi. Alternatively, it may be a more stable compartment, which might act as a sorting station for bidirectional ER-Golgi traffic[2,3]. Using synchronous cargo release, it was shown that ER-to-Golgi trafficking is supported by an intertwined network of tubules, which is dynamic and decorated by both COPII and COPI coats[4]. Parts of the ERGIC may be contiguous with ERES, with segregation of membrane lipids and proteins resulting from effects of membrane curvature, coat proteins, cargo receptors, and other factors[4]. Protein sorting occurs at the ERGIC, so that anterograde and retrograde carriers mediate the further trafficking of proteins and membrane lipids from this compartment[2,3]. In addition, the ERGIC maintains distinct luminal conditions, including calcium levels and pH, setting it apart from the ER and Golgi, and highlighting its unique importance in the secretory pathway. Data support the idea that the ERGIC itself undergoes a maturation process to generate the cis cisterna of the Golgi complex[5]. How the sorting of various proteins at the ERGIC occurs, and how the ERGIC matures to form the cis-Golgi or gives rise to anterograde carriers, is poorly understood.

[1]Section of Endocrinology and Metabolism, Department of Internal Medicine, Yale School of Medicine, New Haven, CT, USA. [2]Department of Cell Biology, Yale School of Medicine, New Haven, CT, USA. [3]Université Paris Cité, CNRS UMR7592, Institut Jacques Monod, Paris, France. [4]Yale Center for Molecular and Systems Metabolism, Yale School of Medicine, New Haven, CT, USA. [5]Present address: Evolution Health Group, LLC, New York, NY, USA. [6]Present address: Eukaryotic Microbiology, Institute of Functional Microbial Genomics, Heinrich-Heine-University, Düsseldorf, Germany. Independent researcher: You Yan. ✉e-mail: anupparchure@gmail.com; jonathan.bogan@yale.edu

In addition to its role in the conventional ER-Golgi secretory pathway, the ERGIC participates in a host of other cellular functions. The ERGIC can detect improperly folded proteins, and serves as a backup system for ER-associated degradation[2]. The ERGIC acts in macroautophagy (hereafter termed 'autophagy') and provides membranes for autophagosome biogenesis[6,7]. Finally, the ERGIC functions in "unconventional" secretion pathways by which specific membrane proteins are delivered to endosomes or directly to the plasma membrane (PM), bypassing the Golgi complex[8,9]. Such unconventional secretion pathways are often cell-type specific and mediate the exocytic translocation of physiologically important membrane proteins to the cell surface. For example, CFTR, a chloride channel that is mutated in cystic fibrosis, traffics to the PM at least in part by using such a Golgi-bypass pathway[10,11]. It remains unknown how membrane trafficking at the ERGIC can be adapted in a cell type-specific manner to control the exocytic translocation of specialized transmembrane cargoes.

The GLUT4 glucose transporter is also proposed to traffic by an unconventional, Golgi-bypass pathway[12–15]. In fat and muscle cells, insulin stimulates glucose uptake by mobilizing GLUT4 to the cell surface. In cells not stimulated with insulin, GLUT4-containing vesicles are trapped near the ERGIC by the action of TUG (Tether, containing a UBX domain, for GLUT4; also called Aspscr1, UBXN9) proteins[16–20]. Insulin triggers site-specific endoproteolytic cleavage of TUG to liberate these vesicles and to load them onto kinesin motors for translocation to the cell surface[21,22]. The TUG C-terminal cleavage product then enters the nucleus and regulates gene expression[23]. Data imply that the insulin-responsive vesicles form, in part, by budding from ERGIC membranes[15,24]. Because TUG localizes at the ERGIC, it is ideally positioned to capture these vesicles and to sequester them away from the PM[18]. Upon insulin stimulation, and in adipocytes with shRNA-mediated TUG depletion, the mobilized vesicles fuse directly at the PM[25]. The precise mechanism by which TUG traps GLUT4-containing vesicles within unstimulated cells remains uncertain. More broadly, GLUT4 expression and TUG cleavage are cell-type-specific processes, observed in fat and muscle but not in other cell types[19,21,23]. TUG itself was discovered >20 years ago and is widely expressed, yet its only well-characterized role is in the trafficking of GLUT4[16,26]. To mediate insulin-responsive GLUT4 trafficking, fat and muscle cells may appropriate a more ubiquitous function of TUG. This function is not understood.

Here, we identify a more ubiquitous regulatory function for TUG. We show that TUG is critical to maintain the ERGIC as a distinct compartment in the early secretory pathway. Cells lacking TUG exhibit a distorted Golgi morphology, together with accelerated flux of a model soluble cargo from the ER to the *cis*-Golgi. TUG contains two intrinsically disordered regions (IDRs) and can form liquid-like biomolecular condensates in vitro. A central IDR is necessary and sufficient for TUG localization to the ERGIC, and an N-terminal domain mediates TUG oligomerization in trans. Finally, cellular processes that rely on the ERGIC are disrupted in TUG knockout cells. The data identify TUG as an essential protein to organize the early secretory pathway and imply that this function is co-opted in specialized cell types to mediate unconventional secretion pathways that bypass the Golgi complex.

## Results

### TUG localizes to the ERGIC and organizes the early secretory pathway

Previous results show that endogenous TUG protein is present both in punctate structures, colocalized with the ERGIC marker ERGIC53, and diffusely throughout the cytosol[16,18]. As well, when an extended linker is used, TUG can be tagged at its C-terminus without disrupting its function in 3T3-L1 adipocytes[21]. Therefore, to image TUG protein in cells without fixation or antibody staining, we used a linker to fuse mCherry fluorescent protein at the TUG C-terminus. In TUG knockout (KO) HeLa cells, this TUG-mCherry protein was enriched in clustered punctate structures that colocalized extensively with

Emerald-tagged ERGIC53[27], and in a diffuse pattern throughout the cytosol and nucleus, recapitulating previous results (Fig. 1a). When Emerald-ERGIC53 was expressed alone, this marker had a typical distribution that partially overlapped with GM130 and with Sec31A, proteins present at the *cis*-Golgi and at ERES, respectively (Supplementary Fig. 1a, b). TUG-mCherry puncta do not overlap with the ERES, as assessed in cells co-expressing Sec23-GFP (Fig. 1b). TUG-mCherry was also excluded from the Golgi apparatus, as shown by co-expression of monomeric Neon Green (mNG) -tagged GMAP210 (Fig. 1c). We observed a similar distribution of TUG-mCherry, with respect to the above markers, when it was expressed in wild-type (WT) HeLa cells (Supplementary Fig. 1c–e). Syntaxin-12 (Stx12, also called Stx13) is consistently associated with GLUT4 in proteomic studies[28–32], and is required for the unconventional secretion of CFTR[11]. We observed partial overlap of TUG-mCherry with GFP-tagged Stx12, (Fig. 1d). These results are expanded upon below. Together, the data are consistent with previous results and show that TUG localizes at the ERGIC in HeLa cells, and that TUG overexpression appears to sequester ERGIC membranes away from the ERES and *cis*-Golgi.

To study the effects of TUG knockout, we used murine embryonic fibroblasts (MEFs). MEFs have ~30-fold greater abundance of TUG protein, compared to HeLa cells (Supplementary Fig. 2a). We previously generated mice containing a conditional knockout allele of TUG (TUG^fl/fl)[23]. Immortalized fibroblasts from WT and TUG^fl/fl embryos were treated with Cre recombinase to generate control and KO MEFs. In WT MEFs, endogenous ERGIC53 was present in both peripheral punctate and Golgi-associated structures (Fig. 2a). In TUG KO MEFs, ERGIC53 was reduced in the periphery, and we observed increased colocalization of ERGIC53 and GM130. To quantify this redistribution, we segmented 3D-confocal stacks to generate ERGIC53 and GM130 surfaces in WT and TUG KO MEFs (Supplementary Fig. 2b). We then compared the number of ERGIC53 structures that did not overlap or touch GM130 structures. As shown in Fig. 2b, there were significantly fewer independent ERGIC53 surfaces, not touching GM130, in TUG KO MEFs, compared to WT control cells ($p < 0.0001$). Conversely, the intensity of ERGIC53 staining in areas overlapping with GM130 was slightly but significantly ($p = 0.0091$) increased in TUG KO MEFs (Fig. 2c). The data support the idea that there is an absorption of ERGIC53 into the *cis*-Golgi in TUG KO MEFs, compared to WT control cells.

The localization of TUG to the ERGIC, together with redistribution of ERGIC53 to the *cis*-Golgi in TUG KO MEFs, led us to consider whether TUG restrains the anterograde flux of soluble cargo from the ER to the Golgi. That is, it may serve as a brake on the early secretory pathway. To test this idea, we used a model cargo that can be pulse-released from the ER. The soluble cargo protein pancreatic adenocarcinoma upregulated factor (PAUF) has previously been used to study protein secretion[33]. PAUF can be tagged with both a fluorescent protein (mKate2) and FM4 domain repeats, and is targeted to the secretory pathway provided that a signal sequence is present at the N-terminus[34,35]. The FM4 domain repeats then cause PAUF aggregation and retention in the ER and permit the triggered release of PAUF into the secretory pathway upon addition of a solubilizing drug (D/D solubilizer)[36,37].

As shown in Fig. 2d, the mKate2-FM4-PAUF reporter aggregated in the ER, marked by protein disulfide isomerase (PDI), and did not overlap with GM130 in the absence of D/D solubilizer, both in WT and TUG KO MEFs. Ten minutes after the addition of solubilizer, most mKate2-FM4-PAUF was concentrated at the *cis*-Golgi in TUG KO MEFs. At the same time point in WT MEFs, there was some signal at the *cis*-Golgi, but also much that remained in the ER. By 15 min, mKate2-FM4-PAUF was concentrated at the *cis*-Golgi in WT cells, and the peripheral ER-accumulated protein had dissipated (Supplementary Fig. 2c). By 25 min after the addition of solubilizer, cargo was present in post-Golgi vesicles, both in WT and TUG KO MEFs (Supplementary Fig. 2c).

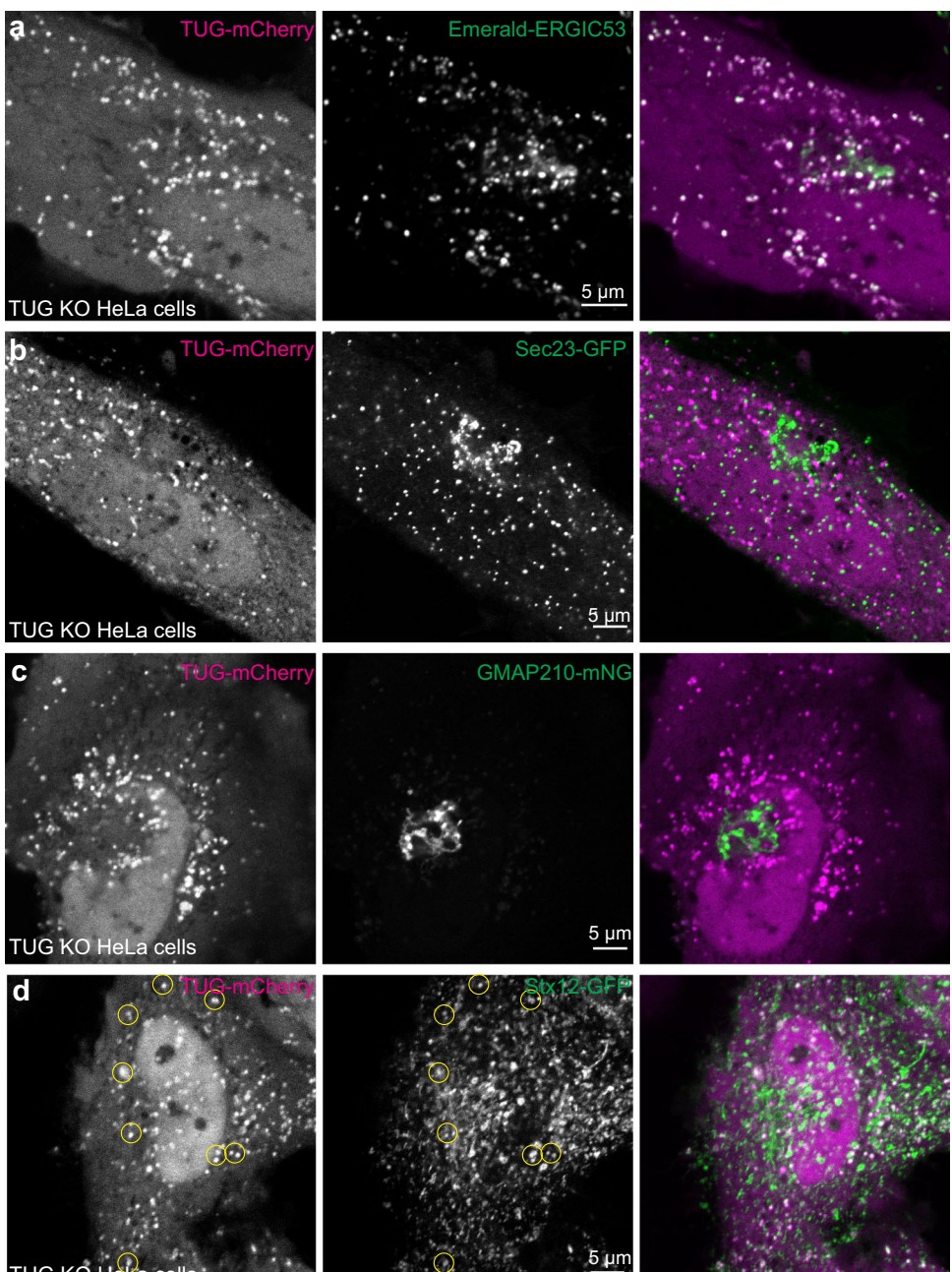

**Fig. 1 | TUG localizes to the ERGIC. a–d** Confocal images are shown of TUG KO HeLa cells co-transfected with mCherry-tagged TUG (Tug-mCherry; magenta) together with Emerald-tagged ERGIC53 (Em-ERGIC53; **a**; green), the ERES marker Sec23-GFP (**b**; green), the Golgi marker mNG-tagged GMAP210 (GMAP210-mNG; **c**; green) and Stx12-GFP (**d**; green). Note the colocalization between TUG and ERGIC53 in peripheral punctate structures. TUG puncta also are positive for Stx12 (yellow circles), but are distinct from the ERES (Sec23) and are excluded from the Golgi apparatus (GMAP210). Images for co-transfection of Tug-mCherry with Em-ERGIC53 are representative of three independent experiments with similar results. Those for Sec23-GFP and GMAP210-mNG and Stx12-GFP are representative of two independent experiments with similar results.

Quantification of the signal enrichment at the Golgi at 10 min after addition of D/D solubilizer confirms a significant increase in TUG KO cells, compared to WT cells (Fig. 2e). The data show that the rate of cargo flux from the ER to the *cis*-Golgi is greater in TUG KO MEFs, compared to WT control cells. The data support the idea that TUG acts as a brake on the transport of a small soluble cargo through the early secretory pathway, which is released in TUG KO cells.

We wondered whether the increased transport kinetics from the ER to the Golgi could be attributed to changes in the distribution of ERES in relation to the *cis*-Golgi. To address this question, we fixed cells that were either i) wildtype (WT), ii) TUG KO and iii) WT MEFs with TUG overexpression, and we labeled them using antibodies to the ERES marker Sec31A and the *cis*-Golgi marker GM130. After segmenting the Golgi and the ERES puncta from confocal images, we counted the number of Sec31A puncta that were touching the Golgi, versus the total number of Sec31A clusters in each cell. We did not observe any changes in clustering of the ERES at the Golgi apparatus in TUG KO MEFs or in MEFs overexpressing TUG, compared to WT control cells (Supplementary Fig. 3a, b). Due to increased compaction of the Golgi in TUG KO cells, described below, there may be an increased density of ERES in proximity to the *cis*-Golgi. We observed no change in the number or percentage of ERES touching the Golgi. Thus, the accelerated flux we

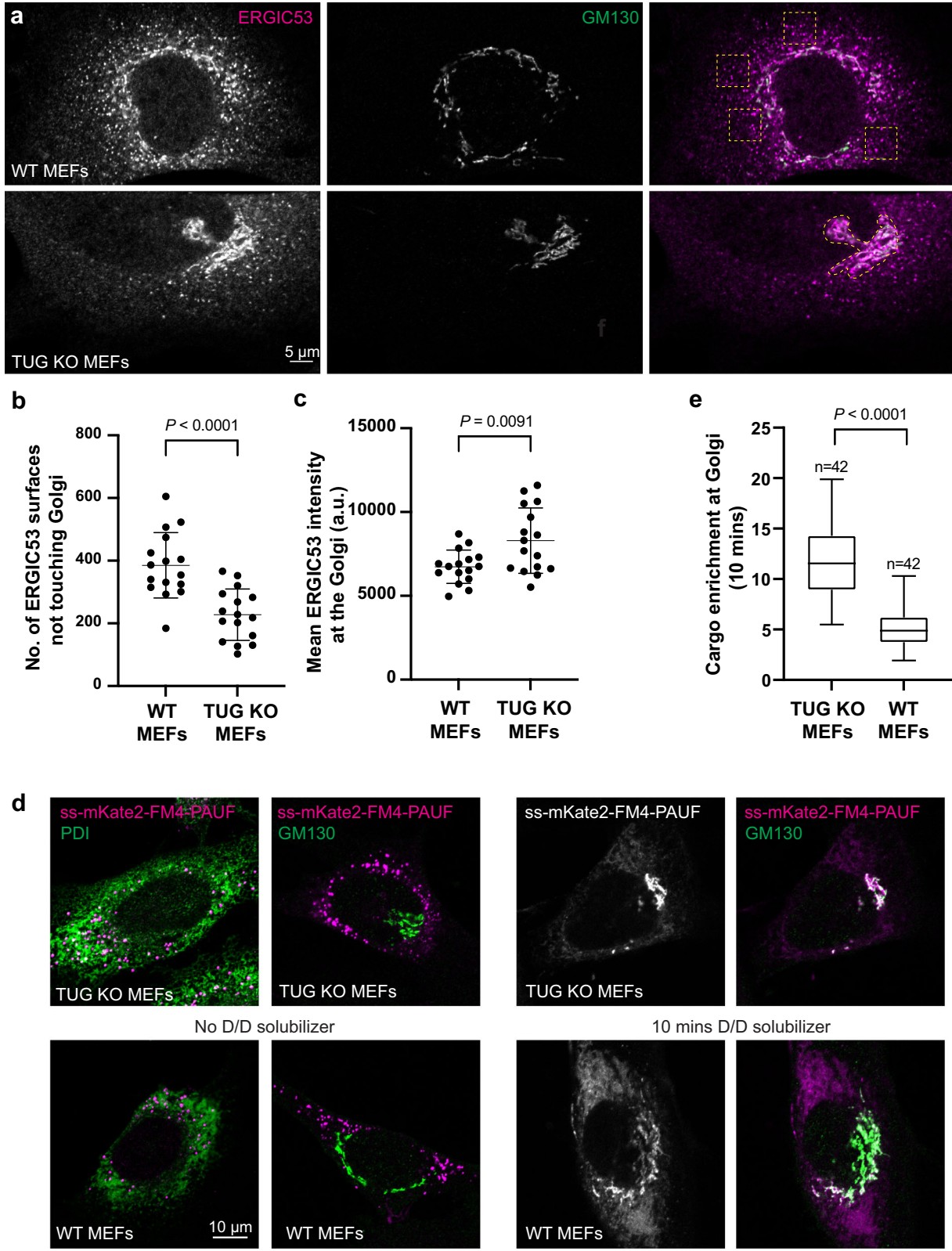

observed cannot be attributed to any large change in ERES distribution relative to the Golgi in TUG KO cells.

## TUG deletion alters Golgi morphology

We hypothesized that in cells lacking TUG, altered trafficking at the ERGIC could result in distorted Golgi morphology. Our previous results showed that siRNA-induced TUG depletion in HeLa cells caused only subtle alterations in Golgi morphology, and brefeldin A removal was required to elicit a robust phenotype[18]. Because TUG is present at greater abundance in MEFs than in HeLa cells (Supplementary Fig. 2a), we considered that the phenotype might be more dramatic in MEFs. Indeed, as assessed by confocal microscopy of two

**Fig. 2 | TUG regulates anterograde trafficking. a** WT (top) and TUG KO (bottom) MEFs were fixed and stained using antibodies to the ERGIC marker ERGIC53 (magenta) and the cis-Golgin GM130 (green). Images are a single slice from a confocal stack. Dashed boxes in the top image denote peripheral punctate staining of ERGIC53, which is devoid of GM130. The dashed region in the bottom image denotes the Golgi, based on GM130 staining. Note the reduction in peripheral ERGIC53 puncta in TUG KO MEFs, compared to WT cells, and increased ERGIC53 distribution at the cis-Golgi. **b** Confocal images were segmented to generate ERGIC53 and GM130 surfaces. The graph compares independent ERGIC53 surfaces (not touching GM130 surfaces) in WT and TUG KO MEFs. A reduction in the number of independent ERGIC53 surfaces was observed in TUG KO MEFs. Data from 16 WT and 16 KO cells are shown; mean ± standard deviation (s.d.). **c** The graph compares average intensities of ERGIC53 staining at the Golgi after collapsing a confocal stack. Data from 16 WT and 16 KO cells are shown; mean ± s.d. **d** Images of TUG KO

(top) and WT (bottom) MEFs containing retroviruses to express PAUF tagged with mKate2 and FM4 domains. The FM4 domains cause aggregation of PAUF in the ER (marked by PDI, in green; left), and is distinct from GM130 (green; right) signal in the absence of D/D solubilizer. The addition of D/D-solubilizer releases PAUF from the ER for anterograde traffic through the secretory pathway. Cells were fixed 10 min after the addition of D/D solubilizer and were imaged (right). In TUG KO MEFs (top), most signal is concentrated at the Golgi, but in WT MEFs there is also surrounding ER signal. **e** Data from replicates of the experiment in **f** were quantified, and enrichment of mKate2-tagged PAUF signal at the Golgi is normalized to the signal in the surrounding ER. $N = 42$ cells in each group, acquired from 4 independent experiments. The box indicates median and quartile ranges, and whiskers show the spread of the data from minimum to maximum. All statistical analyses used an unpaired two-tailed t test with Welch's correction.

cis-Golgi markers, GM130 and the KDEL receptor (KDELr), the Golgi was more compacted in TUG KO MEFs, compared to WT control cells (Fig. 3a). This difference was robust, and we quantified it based on GM130 staining in two different ways. First, the Golgi covered much less of the circumference of the nucleus in TUG KO cells, compared to WT MEFs (Fig. 3b). Second, we characterized the outline of the Golgi itself using the circularity function, which relates the perimeter to the area of the Golgi and returns a number closer to one when the Golgi is more nearly circular. By this measure as well, TUG KO cells have a more compact Golgi morphology (Fig. 3c). Thus, TUG knockout results in a dramatic, compacted Golgi morphology in MEFs, as assessed by confocal microscopy.

To characterize ultrastructural alterations in Golgi morphology in three dimensions in TUG KO MEFs, we used electron microscopy (EM) tomography. This revealed that TUG KO MEFs have both dilated Golgi cisterna and abundant small unfused vesicles surrounding the Golgi stacks (Fig. 3d). The dilated cisterna suggested that there might be an increased distance between cis- and trans- Golgi markers in TUG KO MEFs. Therefore, we also used 4Pi Single Molecule Switching (4Pi-SMS) nanoscopy[38] to image GM130 (cis-Golgi) and Golgin97 (trans-Golgi) in fixed cells (Fig. 3e–g, Supplementary Fig. 4a–d). As predicted, the distance between these markers was greater in TUG KO MEFs than in WT MEFs, consistent with the morphological changes observed by EM tomography.

## TUG protein forms condensates in solution

Analysis of TUG protein in silico using Alphafold[39] and PONDR[40] reveals that along with structured domains (UBL1, UBL2 and UBX), TUG protein contains two predicted intrinsically disordered regions (IDRs) (Supplementary Fig. 5a, b). The larger of these is a central region, IDR1, encompassing residues 183-321 of murine TUG (Supplementary Fig. 5a and Fig. 4a). The second region, IDR2, is at the C-terminus of the protein and includes residues 462-550. IDRs are associated with proteins that form biomolecular condensates and, indeed, TUG is predicted to undergo liquid-like phase separation according to the MolPhase algorithm[41]. To test whether TUG can form condensates in vitro, we expressed an mCherry- and His- tagged TUG protein in BL21 E. coli, then purified this protein using nickel-NTA resin followed by gel exclusion chromatography. Upon buffer exchange to 125 mM NaCl, representing physiological salt concentration, we did not observe any condensate formation by confocal microscopy. However, condensate formation could be induced by the addition of Ficoll 400, a commonly used crowding agent (Supplementary Fig. 5c). This behavior did not depend upon the mCherry tag, as similar results were obtained using mNeon-Green (mNG) -TUG (Supplementary Fig. 5d). We further confirmed that these TUG condensates behaved like liquids. By time lapse imaging, we observed fusion of two or more droplets in proximity, followed by rapid relaxation of the condensate to a spherical shape (Supplementary Fig. 5e). Additionally, after photobleaching a small

area within the droplets, we observed nearly 70% recovery of the fluorescent signal within one minute (Supplementary Fig. 5f, g). Of note, IDR1 is rich in charged, proline, serine, and threonine residues, similar to regions in other proteins that form biomolecular condensates, as discussed below (Supplementary Fig. 5h). Together, the results support that upon molecular crowding, purified TUG protein has the ability to form liquid-like biomolecular condensates.

## The central IDR mediates TUG localization to the ERGIC

To test whether TUG forms condensates in cells, and whether these may act in the early secretory pathway, we used truncated proteins and isolated IDRs fused to mCherry (Fig. 4a). We expressed these proteins in TUG KO HeLa cells and monitored their distribution by confocal microscopy. We reasoned that in cells lacking endogenous TUG, the distribution of ectopically expressed TUG variants would not be influenced by potential effects of oligomerization with endogenous, intact TUG protein. We observed that although full-length TUG had a punctate distribution, proteins lacking the central IDR (TUG-IDR1Δ-mCherry) were present in a diffuse pattern throughout the cytosol (Fig. 4b). Of note, this protein was also completely excluded from the nucleus. We also performed the converse experiment. We fused IDR1 and IDR2 independently with mCherry and expressed these proteins in TUG KO HeLa cells. Although IDR2 was distributed diffusely, IDR1 formed punctate structures in the cytoplasm and was strongly enriched in the nucleus (Fig. 4c). To confirm that the punctate structures of IDR1 were not due to the mCherry tag or the linker sequence in the fusion protein, we generated another fusion construct by appending a monomeric version of superfolder GFP (sfGFP) and a different linker sequence at the C-terminus of IDR1. Expression of sfGFP-tagged IDR1 in TUG KO HeLa cells resulted in similar punctate distribution (Supplementary Fig. 6a). We also confirmed that in TUG KO MEFs, the distributions of both IDR1-mCherry and TUG-IDR1Δ-mCherry were similar to those we observed in HeLa cells (Supplementary Fig. 6b). These results suggested that the central IDR, IDR1, has the potential to form condensates in cells.

To test whether the punctate structures we observed upon expression of IDR1 are condensates, we treated cells with 1,6-hex-anediol, an aliphatic alcohol that has been used to acutely disrupt condensates in cells[42–44]. We did not observe dissolution of IDR1 puncta after hexanediol treatment (Fig. 4d). The spatial distribution of the N-terminal IDR from the FUS protein, a protein which forms biomolecular condensates[45], was distinct from that of TUG-IDR1 (Fig. 4e). Together with data showing that TUG IDR2 is present in a diffuse pattern (Fig. 4c, above), we conclude that the TUG-IDR1 structures are specifically formed upon expression of this polypeptide. This may result from multivalent, dynamic interactions involving charged, proline, and hydroxyl-containing residues in IDR1 (Supplementary Fig. 5h). This would fit with how other intrinsically disordered regions are described to be targeted to specific condensates[46–48].

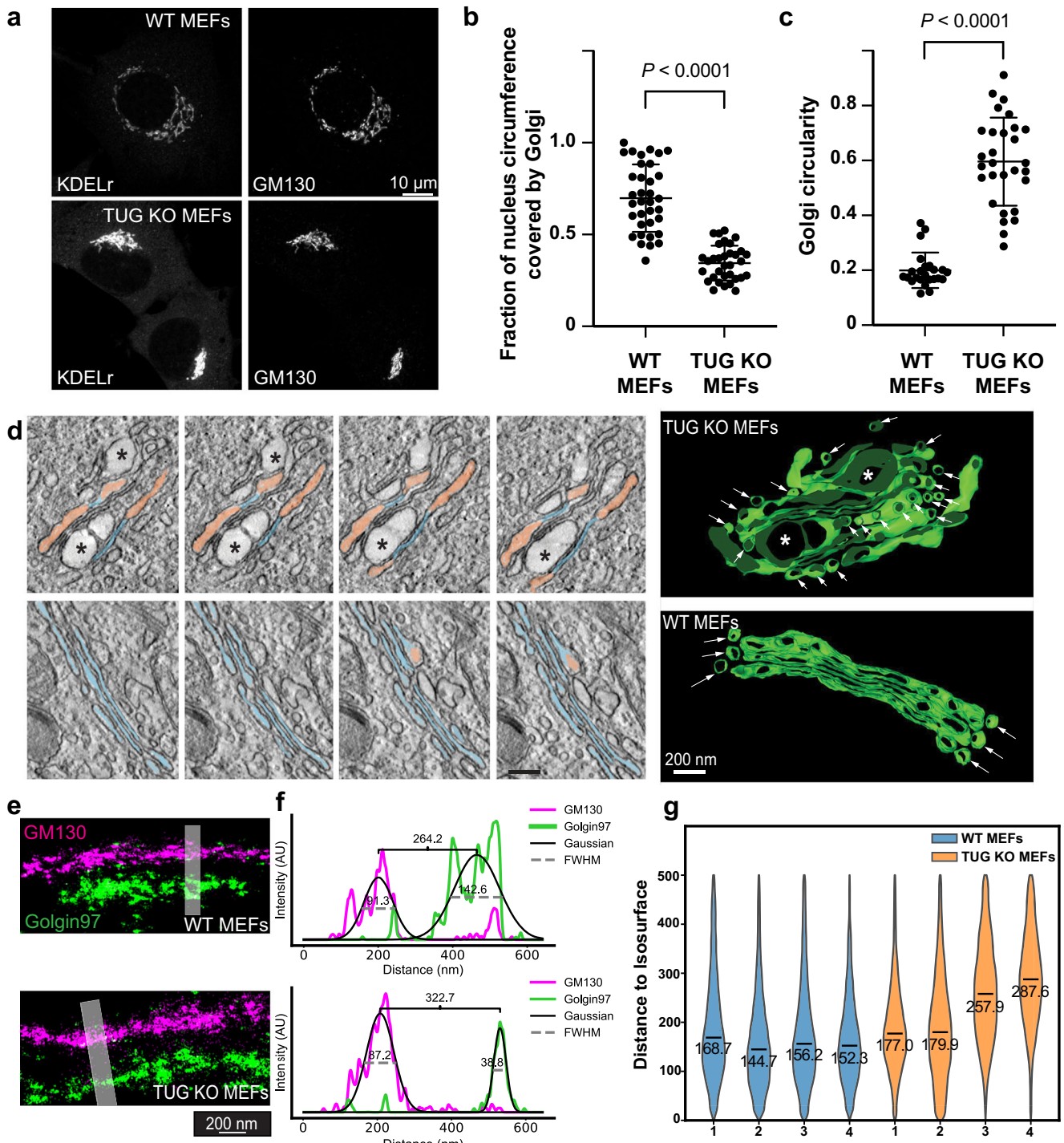

**Fig. 3 | TUG regulates the organization of the Golgi apparatus. a** WT and TUG KO MEFs were immunostained to detect KDELr and GM130. Note the compacted Golgi morphology in TUG KO MEFs. **b** Quantification of Golgi length along the nuclear circumference. Plots show mean ± s.d.; *N* = 35 WT and 33 KO cells, analyzed using a two-tailed *t* test. **c** Compaction of Golgi based on GM130 images was calculated using a circularity function (see Methods). Plots show mean ± s.d.; *N* = 24 WT and 29 KO cells, analyzed using a two-tailed *t* test. **d** Electron microscopy (EM) tomography imaging of Golgi complexes in WT and TUG KO MEFs. At left, cross sections of TUG KO (upper) and WT (lower panels) MEFs are shown. Narrow, well-stacked cisternae are pseudo-colored in blue, dilated cisternae are salmon, and ballooned areas are marked by asterisks. At right, tomographic reconstructions of Golgi complexes in TUG KO and WT MEFs are shown. In addition to dilated cisterna, accumulated small vesicles were observed in TUG KO cells (arrows). **e** 4Pi-SMS side view images of Golgi stacks are shown, immunostained for GM130 (cis-Golgi; magenta) and Golgin97 (trans-Golgi; green). Images from WT and TUG KO MEFs are indicated. Shaded areas were used to quantify the separation of cis and trans markers in line scan profiles. **f** Line scan profiles were generated for GM130 and Golgin97. The intensity profile in each channel was fitted to a gaussian and the peak-to-peak distance was used to measure the separation between these cis and trans markers. In the example shown, this distance was 264 nm in WT MEFs and 323 nm in TUG KO MEFs; the increase in KO cells was ~60 nm. FWHM represents full width at half-maximum. **g** Violin plots of the shortest distances of *trans* points to the *cis* isosurface for four WT and four TUG KO cells (see Supplementary fig. 4). The median value for each cell is indicated. For WT cells, *N* = 254,025, 291,063, 372,879, and 252,424 measurements; for KO cells, *N* = 290,987, 272,770, 156,019, and 163,102 measurements. The overall median distance was 155.0 nm for WT cells and 210.2 nm for KO cells.

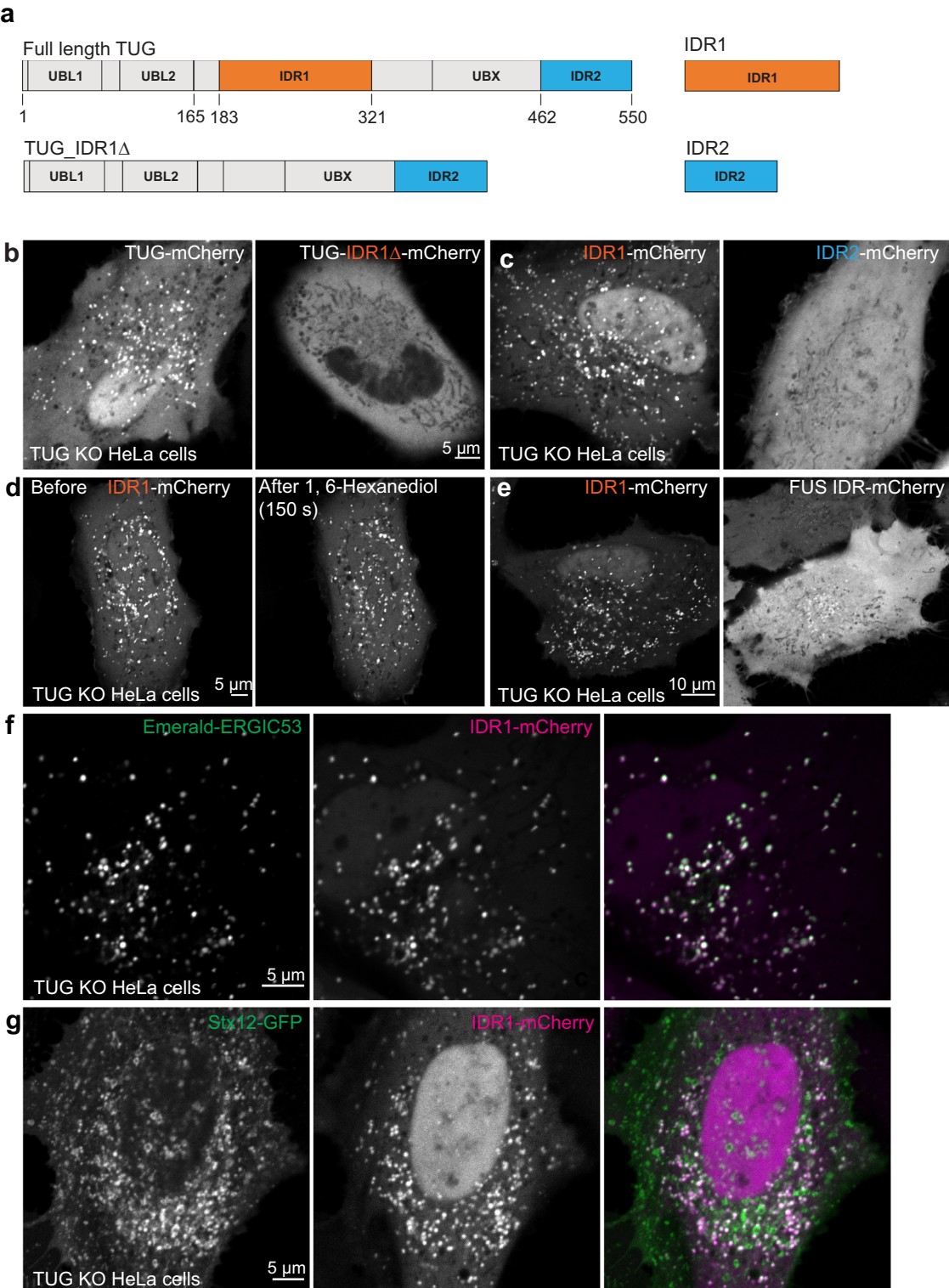

We wondered whether the IDR1 puncta we observed were isolated structures or associated with any cellular organelles. We co-expressed mCherry-tagged IDR1 together with Emerald-tagged EGRIC53, using TUG KO HeLa cells as above, and we observed extensive colocalization of the two proteins (Fig. 4f). This result is similar to that using full-length TUG-mCherry (Fig. 1a, above). In addition, mCherry-tagged IDR1 colocalized with a subset of Stx12 structures in TUG KO HeLa cells (Fig. 4g). This result is also similar to data using full-length TUG (Fig. 1d, above). Furthermore, IDR1 puncta are distinct from the ERES and the *cis*-Golgi, marked

using co-expression of Sec23-GFP and GMAP210-mNG (Supplementary Fig. 6c, d) Together, the data indicate that TUG IDR1 is necessary and sufficient to localize TUG to the ERGIC. To test whether TUG and ERGIC53 interact directly, we expressed Emerald-ERGIC53 in HEK293FT cells and immunoprecipitated ERGIC53 using GFP-trap beads. We did not observe copurification of endogenous TUG protein in the immunoprecipitates, suggesting that these proteins do not interact with high affinity (Supplementary Fig. 6e). We conclude that the central IDR in TUG is unique and contains localization sequences to target TUG to the ERGIC and to the nucleus.

**Fig. 4 | The central disordered region (IDR1) in TUG is necessary and sufficient to mediate TUG localization to the ERGIC. a** Schematic of TUG variants used for expression in TUG KO HeLa cells. All the proteins also contained a C-terminal mCherry tag to aid the visualization of the proteins in living cells. **b** Images of TUG KO HeLa cells transfected with full-length TUG (Tug-mCherry; left) or with a variant where the central IDR, IDR1 was deleted (Tug-IDR1Δ-mCherry; right). Although full-length TUG forms punctate structures, the mutant is present in a diffuse distribution. Also note that the mutant is excluded from the nucleus. **c** Images of TUG KO HeLa cells transfected with IDR1 alone, tagged with mCherry (IDR1-mCherry; left), or with IDR2 alone, also tagged with mCherry (IDR2-mCherry; right). Although IDR1 forms punctate structures in cells, IDR2 is present in a diffuse distribution. In (**b**) and (**c**), a minimum of three independent transfections were performed for Tug-mCherry and IDR1-mCherry, three independent transfections for IDR2-mCherry and two independent transfections for Tug-IDR1Δ-mCherry. **d** Images of TUG KO HeLa cells transfected with IDR1-mCherry and treated with 1,6-hexanediol and imaged before (left) or 150 sec. after 1,6-hexanediol treatment (right). Note that the

punctate structures containing IDR1-mCherry continue to be present after 1,6-hexanediol treatment. The structures before and after are not necessarily at the same position, due to shift and refocusing during imaging after drug treatment. The treatment with 1,6-hexanediol was done three times, with similar results each time. **e** Images of TUG KO HeLa cells transfected with IDR1-mCherry (left) and mCherry-tagged IDR from FUS protein (right). Note the differences in the distribution of the two IDRs in cells. mCherry-tagged FUS IDR was transfected in TUG KO HeLa cells in two independent experiments, with similar results. **f** Images from TUG KO HeLa cells co-transfected with IDR1-mCherry (red) and Em-ERGIC53 (green). Note the colocalization of the two proteins in peripheral punctate structures. **g** Images from TUG KO HeLa cells co-transfected with IDR1-mCherry (red) and Stx12-GFP (green). Note the partial colocalization of the two proteins in punctate structures. Representative images for colocalization of IDR1-mCherry were from three independent experiments with Em-ERGIC53 (**f**) and two independent experiments with Stx12-GFP (**g**).

Based on current and previous data, both of these targeting sequences have important roles in mediating TUG function in cells.

## TUG can oligomerize in trans via N-terminal ubiquitin-like domains

We next sought to test whether TUG is able to drive the recruitment of ERGIC membranes in cells. Previous studies show that ectopic localization of golgin proteins, conferred by a mitochondrial targeting signal, is sufficient to capture specific, Golgi-bound vesicles on mitochondria[49]. We wondered if targeting TUG to mitochondria might similarly capture ERGIC membranes at this ectopic location. We used the mitochondrial outer membrane targeting sequence from monoamine oxidase A (MOA) to target TUG-mCherry to mitochondria, and we expressed this TUG-mCherry-mito protein in TUG KO MEFs. Surprisingly, we observed dramatic clumping of the mitochondria themselves (Fig. 5a), rather than marked recruitment of ERGIC53-labeled membranes. Control experiments in which only mCherry was targeted to mitochondria showed that the mitochondria had a filamentous structure and were dispersed throughout the cells (Fig. 5b). Thus, the mitochondrial clumping was due to TUG itself, not the mCherry tag or targeting signal. When TUG KO MEFs expressing mitochondrially-targeted TUG were examined by electron microscopy, the mitochondria were stacked against each other, distinct from the dispersed mitochondria observed in control cells (Fig. 5c, d). To monitor the nature of these assemblies, we expressed mitochondrially targeted TUG in TUG KO HeLa cells, bleached a small region within the assemblies, and monitored the recovery of mCherry-tagged TUG protein. We observed approximately 30% recovery of fluorescence within a minute after bleaching, with a major fraction of this recovery in the first 5–10 s (Supplementary Fig. 7a, b). The data indicate that TUG protein can oligomerize in trans when targeted to membrane-bound organelles, and that the affinity of these oligomeric complexes is sufficient to drive mitochondrial clumping. Moreover, when tethered to the mitochondria, TUG oligomers display liquid-like behavior.

We reasoned that we could use mitochondrial clumping as an assay to identify regions of TUG that are responsible for its oligomerization. Because we observed TUG condensates in vitro, and because condensate formation can be mediated by interactions among disordered regions, we wondered whether one or both IDRs in TUG would be responsible for the mitochondrial clumping we observed. Accordingly, we expressed a mitochondrially-targeted version of TUG in which both IDRs, IDR1 and IDR2, were deleted. This protein continued to cause clumping of mitochondria, similar to intact TUG (Fig. 5e). Conversely, targeting IDR1 alone to the mitochondria failed to cause clumping (Fig. 5f). When we expressed a mitochondrially-targeted form of a larger fragment of TUG, containing the UBL1, UBL2, and IDR1 domains (residues 1–321), we again observed the clumping of mitochondria (Fig. 5g). Finally, as IDR1 did not appear

to mediate this effect, we expressed a mitochondrially-targeted fragment containing residues 1-183. This fragment was sufficient to mediate mitochondrial clumping (Fig. 5h). This fragment corresponds to the tandem ubiquitin-like (UBL1, UBL2) domains, which are predicted by AlphaFold to extend through residue 173[22].

We next asked whether the N-terminus contributes to in vitro condensation of TUG. We considered that the N-terminus might multimerize and seed TUG condensation. To test this idea, we purified a mCherry-tagged TUG protein in which the first 183 residues, including the UBL1 and UBL2 domains, were deleted. We assessed the ability of this mutant to form condensates in solution, using lower amounts of molecular crowder and at varying protein concentrations. Compared to the wild-type protein under the same conditions, the mutant formed smaller droplets, covering less area in the imaging field of view (Fig. 5i, j). This observation could reflect a decrease in the valency of TUG, which might lead to less condensation[50]. Yet, given cellular and in vitro data supporting the idea that the N-terminal structured region of TUG oligomerizes, the simplest explanation is that this oligomerization promotes larger-scale phase separation.

## TUG regulates autophagy, collagen secretion and CFTR trafficking

We hypothesized that cellular functions that rely on the ERGIC might be disrupted in TUG KO MEFs. The ERGIC is well known to serve as a source of membranes for autophagosome biogenesis[6,7]. Thus, we tested whether autophagy is impaired in TUG KO cells. We used a Halo-GFP-LC3B reporter protein, which is proteolytically processed during autophagy to generate a protease-resistant Halo ligand-bound fluorescent fragment[51]. The relative abundance of this fragment corresponds to the rate of autophagic flux and can be measured using in-gel fluorescence imaging. We expressed this LC3 reporter using retroviruses in WT and TUG KO MEFs, and isolated cells with similar levels of expression using FACS. We then starved cells to induce autophagy and analyzed processing of the reporter, as diagrammed in Supplementary Fig. 8a. As shown in Fig. 6a, the abundance of processed LC3 reporter, observed by in-gel fluorescence, was reduced in TUG KO MEFs, compared to WT control cells. We quantified replicate experiments, which showed that the ratio of processed to unprocessed LC3 reporter was reduced by half in TUG KO cells (Fig. 6b). The ratio of processed to total LC3 reporter, which corresponds to autophagic flux, was also significantly reduced in TUG KO MEFs, compared to WT control MEFs (Fig. 6c). Since this assay relies on an ectopic LC3 reporter, we also probed endogenous LC3B levels upon starvation using confocal imaging. Cells were starved and treated simultaneously with Bafilomycin A1 (BafA1), which blocks autophagosome-lysosome fusion and lysosome acidification. Hence, defective autophagosome biogenesis would result in decreased accumulation of LC3B signal. Consistent with this, we observed a ~35% decrease in LC3B staining in TUG KO

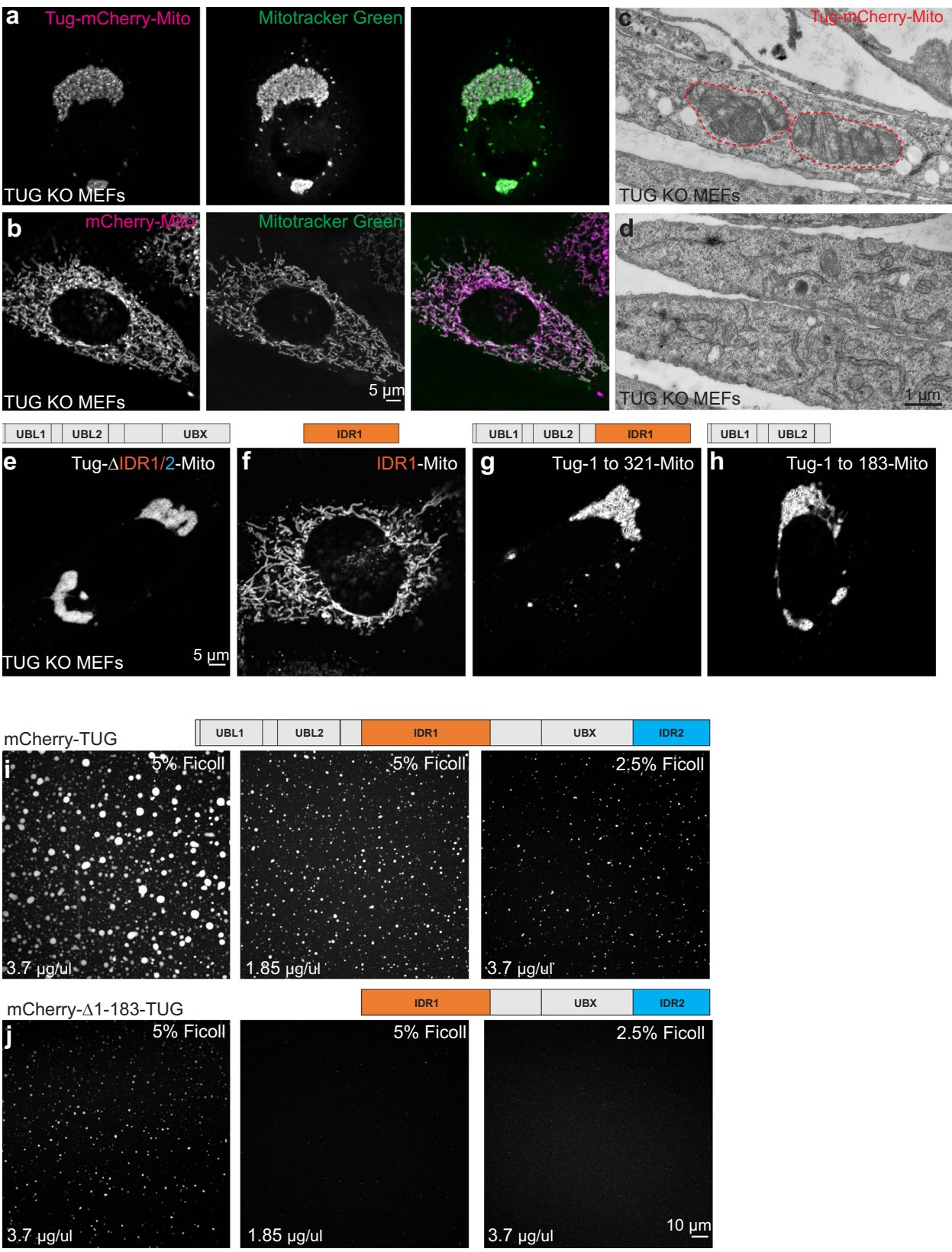

MEFs, compared to WT cells (Fig. 6d, e). Moreover, without inducing starvation, LC3B levels were much lower and were similar in WT and TUG KO cells (Supplementary Fig. 8b). We conclude that TUG deletion causes a reduction in the rate of autophagic flux, consistent with the idea that the ERGIC is regulated by TUG and functions in autophagosome biogenesis.

Large cargoes such as collagens are too big to fit into the small COPII vesicles that typically bud from the ER[52,53]. Previous results show that the ERES resident protein, Tango1, acts together with COPII machinery to build a mega-carrier that can accommodate these large cargoes. Recent data further show that not all ERES are equivalent, and that collagens are exported from a dedicated subset of ERES that may

**Fig. 5 | TUG oligomerizes in trans via its N-terminal ubiquitin-like domains.**
**a** Images of TUG KO MEFs infected with retroviruses to express mCherry-tagged
TUG (magenta) tethered to mitochondria by the fusion of a transmembrane
domain from the protein mitochondrial monoamine oxidase A (Tug-mCherry-
Mito). Living cells were also incubated with Mitotracker green (green), which was
washed off prior to imaging. Tethering of TUG to mitochondria results in mito-
chondrial clumping. **b** Images of TUG KO MEFs expressing mCherry tethered to
mitochondria (magenta), together with Mitotracker green (green). Mitochondria
were present in a filamentous organization under this condition. **c, d** Electron
micrographs from TUG KO MEFs expressing Tug-mCherry-Mito (**c**) or without any
exogenous protein expression (control; **d**) are shown. Mitochondria appear
stacked when TUG is tethered to the mitochondrial outer membrane by mono-
amine oxidase A, but are dispersed in control cells. **e–h** Images of TUG KO MEFs
expressing different truncations of TUG tagged with mCherry and tethered to the
mitochondria. Deletion of IDR1 and IDR2 shows that these regions are not required
for clumping of mitochondria (**e**); IDR1 alone is not sufficient to clump

mitochondria (**f**). The N-terminal structured UBL1 and UBL2 regions in TUG protein
are sufficient to mediate protein-protein interaction in trans and thus to clump
mitochondria when fused to the transmembrane domain of mitochondrial mono-
amine oxidase A (**g, h**). All the above images are from TUG KO MEFs stably
expressing the constructs; cells were imaged in two independent experiments.
**i, j** Purified mCherry-tagged TUG (**i**) and a truncated version (deletion of N-terminal
UBL1 and UBL2 domains, (**j**) were incubated in the presence of different con-
centrations of Ficoll 400 and the condensates were imaged at same settings using
confocal microscopy. Based on the size and intensity of the condensates, it is
evident that the full-length protein is more effective in condensate formation
compared to the deletion mutant, implying the structured UBL domains function
to promote TUG condensation. Images with 5% Ficoll 400 were all acquired at
similar setting for the wildtype and the mutant protein. Images with 2.5% Ficoll are
acquired in the same imaging settings for the wildtype and the mutant protein.
Experiments were repeated at least twice with similar results.

form tunnels capable of transporting collagens from the ER[35]. The
ERGIC provides the membrane necessary to form these large carriers[54].
We reasoned that a defect in membrane supply might cause impaired
collagen export in cells lacking TUG. To study effects on the entire
repertoire of collagens produced in MEFs, we analyzed the secretomes
of WT and TUG KO cells using mass spectrometry. This approach
enabled us to quantify the abundances of individual, secreted proteins
(Supplementary Data 1). We observed that bulky collagens were sig-
nificantly less abundant in the secretome of TUG KO MEFs, compared
to WT control cells (Fig. 6e). As a control, we also examined the
abundances of several small, secreted proteins in the same datasets.
These were not affected by TUG deletion (Fig. 6g). The data demon-
strate that TUG deletion specifically and dramatically reduces the
secretion of bulky collagens, supporting the idea that it acts at the
ERGIC to permit the formation of large carriers required for these
proteins.

Another function of the ERGIC is to support unconventional
secretion, a pathway by which some transmembrane proteins may
bypass the Golgi and traffic directly to the PM[9,55]. We hypothesized
that the trafficking of proteins that participate in such a pathway
might be altered in TUG KO cells. To test this idea, we focused on
CFTR, which has been shown to traffic, at least in part, by a Golgi-
bypass mechanism involving Stx12[10,11]. We expressed GFP-tagged
CFTR[56,57] in WT and TUG KO MEFs and monitored the localization of
this protein using confocal microscopy. As shown in Fig. 6h, GFP-
CFTR protein reached the cell surface in WT cells and did not accu-
mulate in Golgi membranes marked by GM130. In contrast, in TUG
KO MEFs, there was a striking accumulation of GFP-CFTR at the Golgi,
with much less of this protein at the plasma membrane. The overlap
of GFP-CFTR with GM130 was quantified in several cells and is plotted
in Fig. 6i, and showed a 4.6-fold increase in overlap in TUG KO cells
(p < 0.0001). To test whether the arrest in CFTR at the Golgi appa-
ratus was specific or generalized, we generated stable WT and TUG
KO MEFs expressing GFP with a signal sequence and monitored its
secretion. A generalized block in secretion would have affected
secretion of soluble GFP, however we did not notice any differences
between WT and TUG KO cells (Fig. 6j, k). The results support the
view that TUG regulates membrane trafficking at the ERGIC, and that
perturbation of membrane homeostasis in TUG KO cells affects the
trafficking of proteins that participate in a Golgi-bypass mechanism
for trafficking to the PM.

## Discussion

Here we demonstrate that the TUG protein is critical for membrane
homeostasis in the early secretory pathway. Our data support the
concept that the endoplasmic reticulum (ER) -Golgi intermediate
compartment (ERGIC) is a distinct organelle that is organized in part
through the action of TUG. In cells lacking TUG, the ERGIC is absorbed

into the Golgi and the anterograde flux of a model soluble cargo
protein to the Golgi is accelerated. Increased delivery of membranes to
the Golgi can account for the distortion of Golgi architecture we
observe. As well, cellular functions that depend on the ERGIC,
including autophagy, collagen secretion and CFTR targeting, are dis-
rupted. As diagrammed in Supplementary Fig. 9, TUG is an essential
protein to organize the early secretory pathway, which enables it to
serve as a hub for sorting of distinct soluble and transmembrane
protein cargoes and for trafficking of membranes.

TUG may act, in part, as a brake in the early secretory pathway,
which may be cargo selective. We observed increased anterograde flux
of the soluble cargo, PAUF, upon TUG deletion, which may result from
relieving this brake. We also show that an ERGIC marker, ERGIC53, is
absorbed into the *cis*-Golgi in TUG KO cells. Perturbed membrane
homeostasis at the ERGIC may cause the distorted Golgi morphology
we observed. Thus, the effects of TUG KO may be more general. TUG
deletion may cause an increase in the biogenesis of COPII carriers at
the ER, or anterograde carriers from the ERGIC or may enhance
membrane fusion at the ERGIC or *cis*-Golgi. Regardless, TUG acts as a
negative regulator of specific cargoes in the early secretory pathway.

TUG deletion has more marked effects on other ERGIC-dependent
cellular functions. The reduced autophagy we observed in TUG KO
cells may be due, in part, to the absorption of ERGIC membranes into
the *cis*-Golgi. The dramatic impairment of collagen secretion might
seem contradictory at first, yet recent studies show that collagens are
transported from only a subset of ERES, which may give rise to large
carriers[35]. These ERES may be disrupted in TUG KO cells, and desta-
bilization of ERGIC membranes might leave the flux of small COPII-
coated vesicles unaffected or increased. It is also possible that defec-
tive collagen secretion results, in part, from perturbations in Golgi
structure. Similar considerations may apply for CFTR, which uses a
trafficking pathway that is distinct from that used by soluble secretory
proteins[55]. Thus, TUG is a regulator at the ERGIC, which may help
distinguish different types of carriers or sorting pathways at the ER-
Golgi interface.

A central disordered region in TUG is necessary and sufficient for
its recruitment to ERGIC membranes. It remains uncertain how this
recruitment occurs. Our data suggest that TUG does not bind with high
affinity to ERGIC53, yet it may interact with other proteins or mem-
branes present at the ERGIC. Our data also show that TUG can form
biomolecular condensates in vitro. We do not know whether TUG
condensation has a functional role in organizing the ERGIC. Functional
assessment in cells is challenging since the central IDR also mediates
TUG recruitment at the ERGIC. Data suggest that other Golgi proteins
form biomolecular condensates, and GM130 may participate in a
phase-separated structure linking the Golgi ribbon or coating the *cis*-
Golgi cisterna[58–61]. Other proteins at the early secretory pathway,
including TFG and Sec16A, are thought to form biomolecular

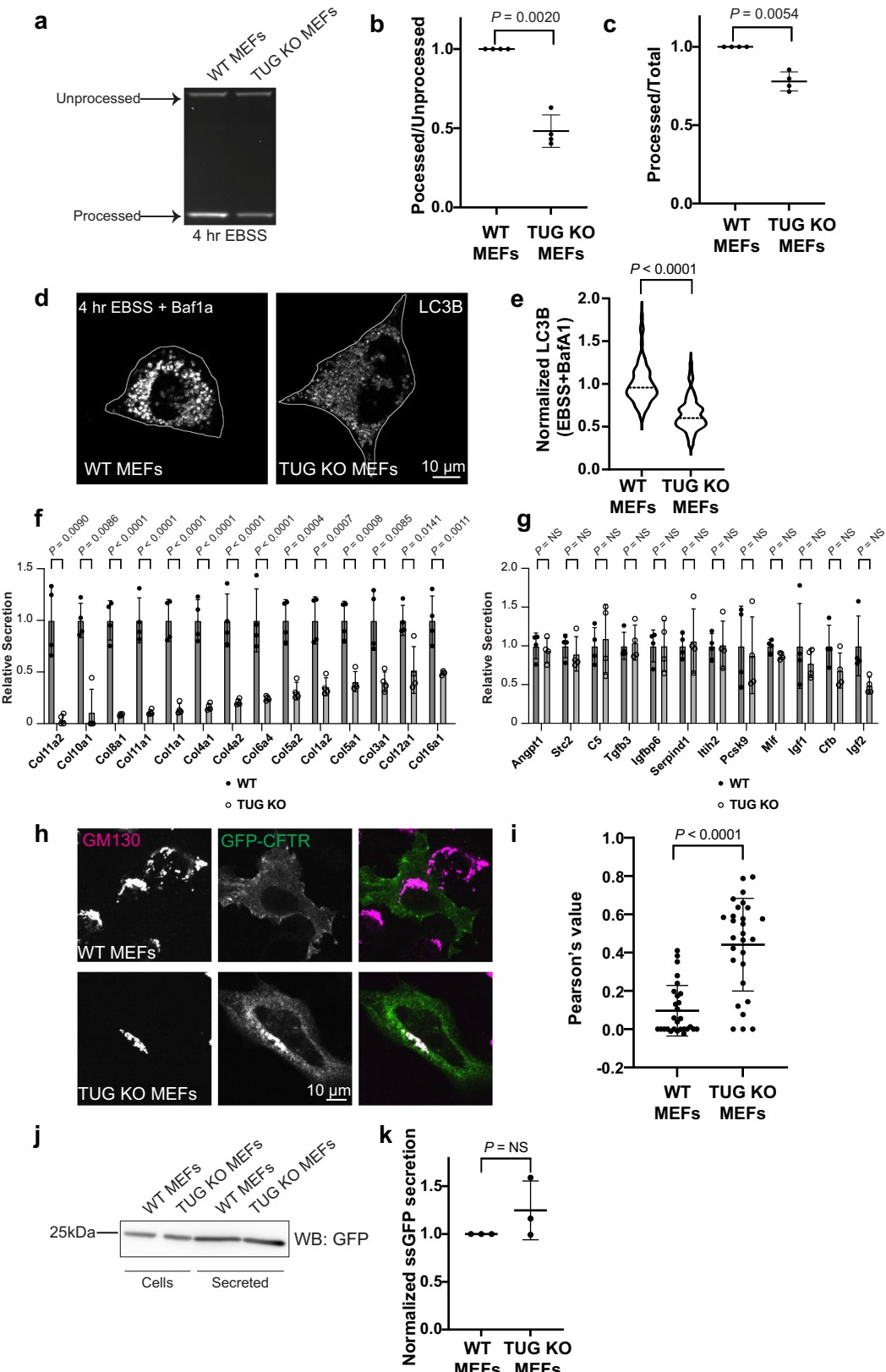

condensates, which may organize membranes and control cargo flux[62-65]. TUG shares some similarity with TFG, since the disordered C-terminus of TFG mediates its recruitment to the early secretory pathway[66]. Our data also show that TUG can oligomerize in trans through its N-terminal ubiquitin-like domains. The deletion of these N-terminal ubiquitin-like domains reduces TUG condensation in vitro,

suggesting a hierarchical organization in building a TUG condensate. Yet, we cannot rule out that the N-terminal ubiquitin-like domains directly mediate TUG condensation, independent of the disordered regions. Further studies will be required to dissect the role of TUG condensation, especially as the N-terminal UBL domains may also regulate other cellular functions.

**Fig. 6 | Physiological effects of TUG deletion on autophagy, collagen secretion and CFTR trafficking. a** In-gel fluorescence was used to monitor autophagic flux in WT and TUG KO MEFs. Processed and processed LC3 reporter bands were quantified. Note the low intensity of the processed band in KO MEFs. **b, c** Quantification of the ratio of processed to unprocessed LC3 bands (**b**) and ratio of intensities of processed to total LC3 reporter (**c**) from $N = 4$ independent experiments, normalized with respect to WT cells. Mean ± s.d., analyzed using a two-tailed $t$ test. **d** Confocal images of WT and TUG KO MEFs, starved for 4 h in EBSS with BafA1 and stained for endogenous LC3B. Note the decreased LC3B in KO cells. **e** LC3B intensities per cell are plotted, normalized to the mean in WT cells. Medians are indicated. $N = 70$ WT and 61 KO cells from two independent experiments, analyzed using a two-tailed $t$ test with Welch's correction. **f, g** Abundances of secreted collagens (**f**) and control proteins (**g**) were quantified using mass spectrometry. Note the decreased collagen secretion in TUG KO MEFs. $N = 4$ independent experiments;

mean ± s.d., analyzed using a two-tailed $t$ test adjusted for multiple comparisons across all secreted proteins. **h** Images from WT and TUG KO MEFs transfected with GFP-tagged CFTR and stained using GFP booster (green) and an antibody to GM130. GFP-CFTR accumulates at the Golgi in TUG KO MEFs. **i** Pearson's correlation coefficient was used to quantify the overlap of GFP-CFTR and GM130. $N = 28$ WT and 28 KO cells from three independent experiments. Mean ± s.d., analyzed by a two-tailed $t$ test with Welch's correction. Note that Pearson's coefficient was increased in TUG KO cell, indicating that CFTR accumulates at the Golgi. **j** Immunoblot to detect abundances of ectopically expressed signal sequence-GFP (ssGFP) in cell lysates and secreted into media in WT and TUG KO MEFs. **k** Plots show the amount of secreted GFP, normalized to that in cell lysates. Data are normalized to WT cells. $N = 3$ independent experiments. Mean ± s.d., analyzed by a two-tailed $t$ test with Welch's correction.

Our studies on TUG localization also illuminate its role in GLUT4 trafficking. Previous results show that in fat and muscle cells, GLUT4 binds directly to TUG and is retained intracellularly by the action of TUG proteins[17,23]. Insulin stimulates the endoproteolytic cleavage of TUG to release this trapped GLUT4 and to mobilize it to the plasma membrane[19,21,67]. Data support the idea that the N-terminal TUG cleavage product, containing the UBL1 and UBL2 domains, is a ubiquitin-like protein modifier, called TUGUL, which is covalently attached to KIF5B motor proteins. Because this N-terminal product binds directly (and noncovalently) to GLUT4 and IRAP (another transmembrane cargo in the GLUT4 vesicles), its attachment to KIF5B can load these vesicles onto kinesin motors for long-range movement to the cell surface[68,69]. Data show that these vesicles fuse directly at the plasma membrane[19,25]. The insulin-responsive vesicles containing GLUT4 are formed, at least in part, by budding at the ERGIC[15,24]. Thus, our present data add further support to the idea that GLUT4 is translocated to the PM by an unconventional, Golgi-bypass pathway[12–14,18]. It is not clear whether binding of GLUT4 and IRAP to the TUG N-terminal ubiquitin-like domains affects the ability of these domains to oligomerize. TUG cleavage and insulin-responsive GLUT4 translocation are cell-type specific. Understanding TUG function at the early secretory pathway will be essential to learning how this general mechanism is adapted to mediate insulin action in fat and muscle cells.

We propose that CFTR and GLUT4 both engage shared membrane trafficking machinery at the ERGIC, and that this machinery may include Stx12 and TUG. Current and previous data show that TUG is required for the proper targeting and retention of CFTR and GLUT4 in the early secretory pathway. Additionally, a subset of Stx12-positive structures overlaps with TUG puncta in cells. Stx12 is described to act at endosomes and other sites[70–74]. Our data suggest a role for Stx12 at the ERGIC, but what trafficking pathway it mediates is not known. Stx12 is required for unconventional secretion of CFTR[11] and is associated with GLUT4 in proteomic studies[28–32]. Whether Stx12 acts in GLUT4 trafficking is not known. The different targeting of CFTR and GLUT4 in TUG KO cells may result from differences in how these proteins interact with shared trafficking machinery at the ERGIC. Further studies will be needed to understand this machinery and to elucidate the possible role of Stx12 at the ERGIC and in Golgi-bypass trafficking.

Our data suggest that the formation of biomolecular condensates, containing TUG and possibly other components, may be important to trap GLUT4-containing vesicles within unstimulated fat and muscle cells. In nerve terminals, condensates containing synapsin are thought to cluster synaptic vesicles[42,75,76]. Synapsin acts with transmembrane proteins, present in the vesicles, to control the clustering of small vesicles[77]. Possibly, TUG might act similarly to oligomerize or to form a condensate, and thus to cluster small, GLUT4-containing vesicles. The interaction of GLUT4 and IRAP with such a condensate may enable the condensate to act as a sponge, holding the vesicles in an insulin-responsive configuration in unstimulated cells. The formation of biomolecular condensates can be promoted by poly(ADP-ribose)[78,79]. The

main cytosolic poly(ADP-ribose) polymerases, PARP5a and PARP5b, bind to the GLUT4 vesicle cargo protein, IRAP, and may poly(ADP-ribosyl)ate TUG[22,80]. Possibly, poly(ADP-ribosyl)ation could then promote TUG oligomerization or condensate formation and thus control the number of insulin-responsive vesicles that are trapped within unstimulated fat and muscle cells.

In conclusion, data here show that TUG resides at the ERGIC and controls diverse membrane trafficking pathways, and that its function is critical to organize the early secretory pathway. The recruitment of TUG to the ERGIC is mediated by an intrinsically disordered region. At the ERGIC, it can oligomerize and may form biomolecular condensates. These biochemical functions enable TUG to act as a brake on the anterograde flux of a model soluble cargo protein. In addition, the action of TUG to organize membranes helps to maintain the ERGIC as a distinct organelle, separate from the Golgi. Accordingly, TUG is required to support ERGIC-dependent cellular functions, including autophagy, collagen secretion and unconventional secretion of transmembrane proteins. Understanding how TUG acts with other machinery to control membrane dynamics will thus have broad implications for understanding normal cell physiology and a range of human diseases.

## Methods
### Cloning and constructs
For cloning, all PCR amplifications were done using Phusion High-Fidelity DNA polymerase, which was obtained from Thermo Fischer Scientific. Final vectors were generated using Gibson assembly (New England Biolabs; NEB). Restriction digestions were carried out using enzymes from NEB. All reactions were carried out using the manufacturer's protocols. Plasmids were sequenced using the Yale Keck DNA Sequencing Core facility.

To generate the full-length TUG protein tagged with mCherry at the C-terminus, we used TUG with a linker sequence containing a TEV cleavage site followed by an AviTag (BirA biotinylation site), described previously[21]. PCR amplification of the TUG sequence and linker was carried out using a priming site in the pBICD2 vector[12,16,81] (Addgene plasmid # 52873; http://n2t.net/addgene:52873; RRID:Addgene_52873). mCherry was amplified using the RINS1 construct (gift from Dr. Dmytro Yushchenko (Addgene plasmid # 107290; http://n2t.net/addgene:107290; RRID: Addgene_107290). The two fragments were then fused using Gibson assembly, and cloned into the double digested (EcoR1, Not1) pB retroviral expression vector[16]. All other truncations of TUG tagged to mCherry were generated using full length protein as a template. To generate TUG IDR1 tagged to sfGFP, sfGFP fragment was amplified using a Chromogranin B tagged to sfGFP construct[82], which was a gift from Dr. Julia von Blume. To generate mCherry-tagged TUG variants that were tethered to the mitochondrial outer membrane, a gene block (Integrated DNA Technologies; IDT) was synthesized to encode the transmembrane domain from the mitochondrial monoamine oxidase A, which was then used as a fragment in Gibson assembly.

To generate the GMAP210 tagged to mNeon green (mNG), plasmids obtained from Dr. James Rothman were used as templates to amplify the coding regions for GMAP210 and mNG, which were then assembled into the pB vector using Gibson assembly. Cloning of the FUS IDR tagged to mCherry was achieved by PCR amplifying a fragment containing the 214 residues corresponding to the FUS IDR from the pcDNA 3.2-FUS-1-526aa-V5, a gift from Dr. Aaron Gitler (Addgene plasmid # 29609; http://n2t.net/addgene:29609; RRID: Addgene_29609) and then inserted into pB vector along with a C-terminal mCherry tag and a linker that was the same used for cloning of IDR1 from TUG.

Cloning of the ssGFP plasmid to monitor GFP secretion was achieved by amplifying the signal sequence and EGFP from a ssGFP-KDEL plasmid (a gift from Dr. James Rothman's laboratory). The GFP-CFTR plasmid was previously described[56,57] and was a gift from Dr. William Guggino. The pEGF-Sec23A plasmid was a gift from Dr. David Stephens (Addgene plasmid # 66609; http://n2t.net/addgene:66609; RRID:Addgene_66609).

For in vitro experiments to express mCherry and 6X His tagged TUG protein, 6X His and mCherry were appended at the N-terminus of murine TUG protein separated by a TEV protease cleavage site which also serves as a linker sequence. Fragments were PCR amplified and fused with the double-digested pET-15B vector (BamH1 and Nco1) using Gibson assembly. Using this as a parent construct, the N-terminal deletion construct was obtained by designing primers to amplify a fragment from amino acid residue 184 in the TUG protein and then fusing this fragment along with another fragment containing the tags with the double-digested pET-15B vector (BamH1 and Nco1) using Gibson assembly. To clone mNeonGreen (mNG) and 6X His tagged TUG protein, the mNG fragment was amplified from the above-described GMAP210-mNG construct and fused with a fragment containing the untagged TUG protein using overlap PCR and cloned into double-digested pET-15B vector (BamH1 and Nco1) using ligation. All oligonucleotides used for cloning are included in Supplementary Data 2.

## Cell culture

HEK293FT (Thermo Fisher Scientific), HeLa cells (Catalog number CCL-2; ATCC) and MEFs were cultured in high glucose DMEM Glutmax (Gibco; 10569044) supplemented with 10% EquaFETAL bioequivalent serum (Atlas Biologicals; EF-0500-A), 100 U/ml penicillin, 100 μg/ml streptomycin and 0.25 μg/mL of Gibco Amphotericin and 2.5 μg/mL plasmocin (Invivogen) (complete medium). Cells were maintained at 37 °C in the presence of 5% carbon dioxide.

To make TUG knockout HeLa cells, CRISPR-Cas9 genome editing was performed essentially as described[83]. Annealed guide RNA oligonucleotides were designed with the help of the CRISPR design tool (http://crispr.mit.edu), cloned into the Bbs1-digested PX459 V2.0 vector, and transformed into Stbl3-competent *E. coli*. After sequencing to confirm successful cloning, 1.0 μg PX459 V2.0 plasmid containing guide RNA was transfected into 100,000 HeLa cells per well in a six-well plate. Cells were selected with 2 μg/ml puromycin for 2 days to kill untransfected cells. After subsequent replating at single cell density, KO clonal cell lines were identified by Western blotting and confirmed by sequencing of PCR-amplified genomic DNA. The oligos to make the gRNAs to knockout TUG were 5′-caccgCGTGTACACGCAGACTGGGG-3′ and 5′-aaacCCCCAGTCTGCGTGTACACGc-3′. The primers used for PCR to verify knockout were 5′- TGATGGTTTCTTTCCTCTCCTC-3′ and 5′- GGACAGCAGATTTTCCAGTTG-3′.

To generate TUG knockout MEFs, primary cultures of murine embryonic fibroblasts from control mice or mice homozygous for a floxed TUG allele, TUG^fl/fl, were isolated at embryonic day 13.5, using methods described previously[84]. The TUG^fl/fl mice were described previously[23]. Mice were backcrossed to C57BL/6J for several generations prior to isolation of MEFs. Control and floxed MEFs were treated

with Ad5CMVCre (Ad-Cre), an adenovirus containing Cre recombinase, which was purchased from the Gene Transfer Vector Core at the University of Iowa. Controls included TUG^fl/fl cells not treated with Ad-Cre and WT cells exposed to Ad-Cre. Cells were immortalized by multiple passaging.

## Transfection

Transfection of HeLa cells was done using FuGENE HD transfection reagent (Promega Corp.) as per the manufacturer's protocol. Briefly, cells were plated in glass-bottom imaging dishes from Cellvis (D35-14-1.5-N) and transfection was done at approximately 50% confluency. For co-transfection of pmEM-ERGIC53 (gift from Dr. Ke Xu (Addgene plasmid # 170717; http://n2t.net/addgene:170717; RRID: Addgene_170717)), rat Stx12-GFP (with eGFP fused at the C-terminus) or EGFP-Sec23A together with Tug-mCherry or IDR1-mCherry, the ratio of the two DNAs was 1:5 (with the mCherry-tagged constructs, which were in the pB retrovirus vector, being used in higher amounts). Tug-mCherry or IDR1-mCherry and GMAP210-mNG plasmids were transfected in 1:1 ratio. In all transfections, 2.5 μl of transfection reagent was used for 1 μg of DNA, and transfection complexes were formed by incubation in Opti-MEM (Gibco) for 20 min, following which these were added to cells in a dropwise fashion. The amounts of transfection reagent used were scaled according to the final amount of DNA. Cells were maintained in high-glucose DMEM Glutamax complete medium without antibiotics for 48 h prior to imaging.

For imaging experiments using transfected MEFs, cells (WT or TUG KO, ~6500 cells) were seeded in 35 mm glass-bottom dishes (Mattek P35G-1.5-14-C) the day before transfection. MEFs were transfected with 1 μg GFP-CFTR[56] using 2.5 μl FuGENE HD transfection reagent (Promega Corp.).

## Antibody staining

The same protocol was utilized for staining of MEFs and HeLa cells in all experiments, except for those involving imaging of GFP-CFTR together with GM130. Cells were fixed using 4% paraformaldehyde (PFA; Electron Microscopy Sciences) for 20 min. PFA solution was made in 1X PHEM buffer (60 mM PIPES, 25 mM HEPES, 10 mM EGTA, and 4 mM MgSO4·7H$_2$0). Cells were then permeabilized for 5 min in PHEM buffer containing 0.3% NP-40 and 0.05% Triton X-100. After permeabilization, cells were blocked in PHEM buffer containing 0.05% NP-40, 0.05% Triton X-100 and 5% normal goat serum (Jackson Immunoresearch). Primary antibodies (1:300 dilution; overnight incubation) and secondary antibodies (1:500 dilution; one hour incubation) were diluted in blocking buffer. Cells were finally washed in 1X phosphate-buffered saline (PBS; Gibco) and stored in PBS at 4 °C before imaging.

To image samples using 4Pi-SMS nanoscopy WT and TUG KO MEFs were seeded on 30 mm diameter No. 1.5H round coverslips (Thorlabs) and grown for 1-2 days before fixation by 4% PFA for 15 min and permeabilization by 0.3% NP-40, 0.05%Triton X-100 for 3 min. Cells were processed as described above, except the secondary antibodies were incubated for 2 h at room temperature. Primary antibodies were used at 1:1000 dilution and secondary antibodies were used at 1:500 dilution. After antibody incubation, samples were post-fixed in 3% PFA + 0.1% glutaraldehyde for 10 min and stored in PBS at 4 °C.

To monitor the localization of GFP-CFTR in WT and TUG KO MEFs, cells were fixed two days after transfection using 10% neutral buffered formalin solution (Sigma) for 10 min. After fixing, cells were washed with PBS and permeabilized with 0.1% Triton X-100 for 10 min. Cells were then blocked in 5% bovine serum albumin (BSA) in PBS with 0.1% Tween 20 (PBST) for 30 min. Following blocking, cells were incubated with the primary antibody GM130 (BD Biosciences) in blocking buffer (5% BSA/PBST) for 1 h at room temperature (RT). Samples were then

washed three times with PBST and incubated with the secondary antibody, Atto488-labeled GFP-booster (1:200 dilution; Proteintech; gba488), and Hoechst 33342 in blocking buffer for 30 min at RT. After three washes with PBS, samples were stored in PBS at 4 °C before imaging.

## Western blotting

For western blotting, cell lysates were prepared in cold 1% NP40 buffer containing complete protease inhibitor cocktail (Roche). Samples were lysed on ice for 30 min followed by centrifugation at 4 °C at 20,000 × g for 10 min. Pellets were discarded, and cell lysate supernatants were stored at −20 °C till further use. Protein estimation was done using Bradford assay (Biorad). Samples were boiled at 95 °C for 5 min in NuPAGE LDS sample buffer (Invitrogen) containing 3.75 % beta-mercaptoethanol (BME). Samples were electrophoresed using 4%-12% gradient NuPAGE gels (Invitrogen) and NuPAGE MOPS SDS running buffer (Invitrogen). Proteins were transferred from the gels to a nitrocellulose membrane (Biorad) using wet blot system (Invitrogen) and NuPAGE transfer buffer (Invitrogen) containing 10% methanol at 10 volts for 90 min. After transfer to the membranes, they were incubated with 5% milk made in PBS containing 0.1% Tween20 (PBST) for at least one hour. Membranes were incubated with specific primary (1:1000 dilution; overnight incubation) and HRP-coupled secondary antibodies (1:10,000 dilution; one hour incubation). Proteins were detected using chemiluminescence (Pierce ECL; Thermo Fisher Scientific) and imaged on ImageQuant LAS400 (Amersham).

For re-probing the membranes using a different primary antibody, membranes were stripped using the Restore PLUS western blot stripping buffer (Thermo Fischer Scientific) for 15 min at room temperature. The membranes were washed in PBST and blocked and re-probed using a different antibody and processes further as described above.

## Electron microscopy

Cells cultured in 10 cm dishes were fixed in 2.5% glutaraldehyde in 0.1 M sodium cacodylate buffer (pH 7.4) for 1 h at room temperature. After rinsing with buffer, they were scraped in 1% gelatin and spun down in 2% agar to form pellets. Samples were post-fixed in 1% osmium tetroxide for 1 h, dehydrated in a series of ethanol up to 100%, then infiltrated and embedded in Embed 812 medium (Electron Microscopy Sciences). The blocks were cured in 60 °C oven overnight. Thin sections (60 nm) were cut using a Leica ultramicrotome (UC7) and post-stained with 2% uranyl acetate and lead citrate. Sections were examined with a FEI Tecnai transmission electron microscope at 80 kV accelerating voltage, and digital images were recorded with an Olympus Morada CCD camera and iTEM imaging software.

For electron microscopy tomography imaging, 250 nm thick sections were cut using a Leica ultramicrotome, collected on formvar/carbon-coated copper grids, and stained with 2% aqueous uranyl acetate followed by lead citrate. 10 nm PGA gold particles were placed on both sides of the grids as fiducial markers before imaging. The tilt-series (single-axis) were collected using a FEI Tecnai F20 TEM at the accelerating voltage of 200 kV; the tilting range was from −60° to 60° in 1° increments. A FEI Eagle CCD camera (4k × 4k) and SerialEM software were used to collect datasets. Image alignment and 3D reconstruction were performed using IMOD software[85] and manual tracing of membrane contours.

## 4Pi-SMS imaging

Two-color 4Pi-SMS imaging was done on a custom-build microscope[38]. Sample mounting, image acquisition, and data processing were performed as described in a previous publication[86], with modifications to the imaging protocol involving changes in acquisition speed and drift correction.

## Retrovirus production

Retroviruses were generated using HEK293FT cells. HEK293FT cells were plated on poly-Lysine (Millipore Sigma) -coated 10 cm² dishes and grown in high glucose DMEM Glutamax complete medium. When the cells were 80% confluent, they were transfected with a plasmid cocktail containing the retroviral vector (containing the gene of interest) and the packaging plasmid, pCL-Eco[87], using Lipofectamine 2000 (Invitrogen) according to the manufacturer's protocol. 5 μg of DNA for each of the plasmids and 40 μl of Lipofectamine 2000 was used for transfection. On the next day, the medium was replaced with fresh complete medium. 48 h after transfection, media containing virus particles was collected, and passed through a 0.45 μm filter and stored at 4 °C. HEK293FT cells were again supplemented fresh complete medium for one more cycle of virus collection. 72 h after transfection, media containing virus particles was again collected, as above. The 48 h and 72 h collections were pooled and either used immediately for infection or aliquoted and stored at −80 °C.

## ER to Golgi transport assays using mKate2-tagged PAUF

WT and TUG KO MEFs (10,000 cells) were plated on glass bottom imaging dishes from Cellvis (D35-14-1.5-N). The next day, cells were infected with retroviruses to drive the expression of mKate2- and FM4-tagged PAUF using 8ug/ml polybrene (Millipore Sigma). The retroviral expression plasmid, pCX4-ss-mKate2-FM4-PAUF, was a gift from Dr. Yuichi Wakana. Virus-containing medium was removed and replaced with high-glucose DMEM Glutamax complete medium after 24 h of infection. Two days after infection, cells were incubated with 1 μM D/D-solubilizer (Takara; 635054) in high glucose DMEM Glutamax complete medium for different time intervals at 37 °C, then fixed at RT with 4% PFA in PHEM buffer for 20 min. Fixed cells were stained using GM130 antibody (BD Biosciences; 610822), Atto 594 conjugated RFP booster (Proteintech; rba594) to amplify the mKate2 signal and Alexa 488 labeled goat-anti mouse secondary antibody to detect GM130. To monitor the ER retained pool, cells were fixed and processed without addition of the D/D-solubilizer as described above. Imaging was carried out on Zeiss 880 using 63x/1.4 oil objective at room temperature.

## Autophagy flux assays

WT and TUG KO MEFs were infected with retroviruses to drive the expression of Halo- and GFP-tagged LC3. The retroviral expression plasmid was a gift from Dr. Thomas Melia. Cells stably expressing similar amounts of the proteins were FACS-sorted using GFP fluorescence. A day before the assay, 0.5 million cells were plated in a 6 well plate. The next day, cells were incubated with Halo-TMR (100 nM; Promega) for 20 min. Cells were washed twice in PBS and then incubated in Earle's Balanced Salt Solution (EBSS; Gibco) for 4 h at 37 °C to induce autophagy. For control samples, after the TMR-Halo incubation, cells were washed twice in PBS and harvested as described below. Cells were scraped in ice-cold PBS. Cells were centrifuged for 2 min at 2000 × g at 4 °C. The supernatant was discarded, and the cell pellet was lysed in 1% NP40 buffer containing cOmplete protease inhibitor cocktail (Roche). Cells were lysed on ice for 30 min, then lysates were centrifuged at 20,000 × g for 10 min at 4 °C. Pellets were discarded and cell lysate supernatants were stored at −20 °C till further use. Protein content was determined from cell lysates using a Bradford protein assay. Equal amounts of each cell lysate sample were heated in NuPAGE LDS sample buffer (Invitrogen) containing BME for 5 min at 95 °C. Samples were electrophoresed on 4%-12% gradient NuPAGE gels (Invitrogen) using NuPAGE MPOS SDS running buffer (Invitrogen). After separation of proteins, the gels were transferred to MilliQ water and imaged on ChemiDoc MP Imaging system (Biorad). Autophagic flux was quantified by measuring the band intensities of the processed and the unprocessed bands and represented as a ratio of processed to total (processed+unprocessed) for each condition.

## LC3B accumulation

WT and TUG KO MEFs were plated and starved for 4 h in EBSS after washing twice in 1X PBS in the presence of Bafilomycin A1 (100 nM, Enzo Life Sciences, Cat. No. BML-CM110). Cells were then fixed in chilled 100% methanol at −20 °C for 10 min. Post fixation cells were blocked in PHEM buffer 5% normal goat serum for at least 30 min and stained using antibodies against LC3A/B (Cell Signaling Technology, Cat. No. 12741S) and PDI (Enzo Life Sciences, clone 1D3, Cat. No. ADI-SPA-891). Untreated control cells were similarly processed. Cells were then imaged on Zeiss 880 using ×63/1.4 oil objective at room temperature. Average intensity of LC3B per cell was quantified from a single plane from WT and TUG KO cells.

## Secretome analysis using mass spectrometry

Approximately, one million WT and TUG KO MEFs were seeded in each 10 cm tissue culture dish. A day before the experiments, cells were switched into phenol red-free DMEM (Gibco; 21063029) containing 10% EquaFETAL bioequivalent serum and sodium pyruvate (Gibco). Approximately, 36 h after plating, cells were washed three times in serum and phenol red-free DMEM and incubated in 6 ml of the same medium for 4 h. Medium was collected, and cell lysates were prepared using 0.5% NP40-containing buffer for parallel mass spectrometry analyses. Medium was subjected to centrifugation at 1400 rpm at 4 °C for 10 min followed by filtration using a 0.22 μ filter. Filtered medium was then concentrated using centrifugal filter units (Millipore; UFC800324) with a 3 kDa cut off. Approximately, 200 μl of concentrated medium was flash frozen using liquid nitrogen and stored at −80 °C until further use.

Mass spectrometry grade chemicals were used, including acetonitrile (ACN), $H_2O$, ammonium bicarbonate ($NH_4HCO_3$), formic acid (FA), Tris(2-Carboxyethyl) Phosphine Hydrochloride (TCEP-HCl) and S-Methyl methanethiosulphonate (MMTS) (ThermoFisher Scientific). Sequencing grade Trypsin/Lys C mix was from Promega. Ammonium bicarbonate ($NH_4HCO_3$) was from Sigma-Aldrich. For sample preparation, a six-fold volume of cold acetone (−20 °C) was added to each sample volume containing 10 μg of protein extracts. Vortexed tubes were incubated overnight at −20 °C then centrifuged for 10 min at 11,000 rpm at 4 °C. Supernatant was removed, then the protein pellets were dissolved in 8 M urea, 25 mM $NH_4HCO_3$ buffer. Samples were then reduced with 10 mM TCEP-HCl and alkylated with 20 mM MMTS. After a 16-fold dilution in $NH_4HCO_3$, samples were digested overnight at 37 °C by a mixture of trypsin/Lys C (1/20 Enzyme/Substrate ratio). The digested peptides were loaded and desalted on Evotips Pure, provided by Evosep one (Odense, Denmark) according to the manufacturer's procedure.

For LC-MS/MS acquisition, samples were analyzed on a timsTOF Pro 2 mass spectrometer (Bruker Daltonics, Bremen, Germany) coupled to an Evosep one system (Evosep, Odense, Denmark) operating with the 30SPD method developed by the manufacturer. Briefly, the method is based on a 44-min gradient and a total cycle time of 48 min with a C18 analytical column (0.15 × 150 mm, 1.9 μm beads, ref EV-1106) equilibrated at 40 °C and operated at a flow rate of 500 nL/min. $H_2O$/ 0.1 % FA was used as solvent A and ACN/ 0.1 % FA as solvent B. The timsTOF Pro 2 was operated with a DIA-PASEF method comprising 12 pydiAID frames with 3 mass windows per frame resulting in a cycle time of 0.975 s as described in Bruker application note LCMS 218. A total of eight samples were analyzed per study, comprising four secretomes from WT MEFs and four from TUG KO MEFs. Data were also collected from four cell lysates from WT MEFs and four from TUG KO MEFs.

Raw MS data files were processed using Spectronaut 18 (Biognosys, Switzerland). Data were searched against the SwissProt Mus Musculus (06 2024, 17212 entries) + Bovine (06 2024, 6046 entries) databases. Contaminating proteins identified only in the Bovine taxonomy were excluded from the final protein list. Specific tryptic

cleavages were selected and a maximum of 2 missed cleavages were allowed. The following post-translational modifications were considered for identification: Acetyl (Protein N-term), Oxidation (M), Deamidation (NQ) as variable and MMTS (C) as fixed. The maximum number of variable modifications was set to 3. Identifications were filtered based on a 1% precursor and protein Q-value cutoff threshold. The protein LFQ method was set to automatic, and the quantity was set at the MS2 level with a cross-run normalization applied. Multivariate statistics on protein measurements were performed using Qlucore Omics Explorer 3.9 (Qlucore AB, Lund, Sweden). A positive threshold value of 1 was set to allow a log2 transformation of abundance data for normalization, i.e. all abundance data values below the threshold are replaced by 1 before transformation. The transformed data were finally used for statistical analysis, i.e. the evaluation of differentially present proteins between two groups using a two-sided Student's $t$ test. An adjusted $p < 0.05$ was used to filter differential candidates.

## Immunoprecipitation

One million HEK293FT cells were plated in a 6 well plate which was coated with poly-Lysine. Cells were transfected with 1 μg of pmEM-ERGIC53 and 2.5 μl of Lipofectamine 2000. The following day, cells were immunoprecipitated using GFP trap agarose beads (Proteintech; gta) according to the manufacturer's protocols. Briefly cells were lysed in 0.5% NP40 containing buffer and cells were spun down at > 20,000 × $g$ for 10 min at 4 °C. The cell lysate supernatant fraction was then incubated with GFP trap agarose beads which were washed and equilibrated in the dilution buffer (10 mM Tris, 150 mM NaCl, 0.5 mM EDTA) and then incubated with the diluted cell lysate and rotated end-over-end for 1 h at 4 °C. 50 μl of diluted cell lysate was kept aside as input fraction before incubation with the beads. After 1 h incubation, beads were washed three times and resuspended in 2X NuPAGE LDS sample buffer with BME and boiled at 95 °C for 5 mins. Samples were then spun down at 20,000 × $g$ for 10 min and the supernatant was analyzed by SDS-PAGE.

## Antisera

Antibodies used for immunofluorescence were as follows: GM130 (BD Biosciences; 610822), Sec31A (BD Biosciences; 612350), ERGIC53 (Sigma; E1031), Golgin 97 (Proteintech; 12640-1-AP), P230 (Invitrogen; PA5-87716), LC3A/B (Cell Signalling Technology; D3U4C) Atto 488 conjugated GFP booster (Proteintech; gba488), Atto 595 conjugated RFP booster (Proteintech; rba594). Antibody against KDELr was a kind gift from Dr. James Rothman's laboratory. Fluorophore-conjugated secondary antibodies (Alexa 488 and Alexa 568) were obtained from Invitrogen. For 4Pi-SMS imaging VHH anti-mouse CF660C (Biotium) and VHH anti-rabbit AF647 (Jackson Immunoresearch) were used. For western blotting, antibodies directed to GFP (Proteintech; P42212) and GAPDH (Millipore Sigma; MAB374) were commercially obtained. The TUG antibody was described previously[16,20] and is directed to the TUG C-terminal peptide, which is identical in mice and humans. HRP-conjugated secondary antibodies were obtained from Millipore Sigma. Antibody against Sec31A to detect ERES in MEFs was raised in rabbit and is a gift from Dr. Fred Gorelick.

## Live cell imaging

Prior to imaging living cells for the purposes of monitoring the distribution of proteins, cells were incubated in phenol red-free DMEM (Gibco; 21063029) containing 10% EquaFETAL bioequivalent serum. Cells were imaged rapidly on Zeiss 880 using ×63/1.4 oil objective at room temperature. To image mitochondria, cells were labeled with 200 nM Mitotracker green FM (Cell Signaling Technology) for 10 min in phenol red-free DMEM containing 10% EquaFETAL bioequivalent serum. Cells were washed twice using this medium before imaging on a

Zeiss 880 using ×63/1.4 oil objective at room temperature. To monitor the sensitivity of IDR1 puncta to 1,6-Hexanediol, cells were imaged at 37 °C in the presence of 5% carbon dioxide on a Zeiss 880 using ×63/1.4 oil objective. Cells were imaged without the addition of 1,6-Hexanediol (Millipore Sigma; 240117), then an equal volume of medium containing 10% 1,6-Hexanediol was added to the dishes so as to attain a final concentration of 5% for 150 sec. Cells were then reimaged as above.

## GFP-CFTR localization assay
WT and TUG KO MEFs were fixed and stained as described above and imaged using an Andor Dragonfly Spinning Disc Confocal Microscope (Oxford Instruments) equipped with 405-, 488-, and 640-nm laser lines, 60x UPLSAPO 1.4NA silicone oil objective, and Sona sCMOS camera with a 6.5 μm pixel size.

## GFP secretion assay
WT and TUG KO MEFs were infected with retroviruses to express GFP with a signal sequence so as to target it to the secretory pathway. Approximately, 1.2 million cells were plated in a 10 cm tissue culture dish overnight. On the day of the assays, cells were incubated in 5 ml of complete medium and cells were allowed to secrete GFP for 4 h. After this the medium was concentrated using centrifugal filter units (Millipore) with a 3 kDa cut off. Concentrated medium was then incubated with GFP trap agarose beads (Proteintech; gta) for 2 h. GFP trap beads were washed in PBS and preincubated in complete medium for at least 45 min before the addition of concentrated medium to capture secreted GFP. After 2 h of incubation the beads were washed, and the protein was eluted using 2X NuPAGE LDS sample buffer with BME and boiled at 95 °C for 5 min. Samples were then centrifuged at $20,000 \times g$ for 10 min and the supernatant was analyzed by SDS-PAGE. Cell lysates were prepared in cold 1% NP40 buffer containing complete protease inhibitor cocktail (Roche). Samples were lysed on ice for 30 min followed by centrifugation at 4 °C at $20,000 \times g$ for 10 min. Pellets were discarded, and cell lysate supernatants were stored at −20 °C till further use. To analyze relative secretion, cell lysates and IP samples were analyzed using western blotting and probed using antibody directed against GFP and the band intensities in the IP fraction were normalized to that in cell lysates for WT and TUG KO conditions.

## Image analysis
To quantify Golgi morphology, WT and TUG KO MEFs were labeled using antibodies to GM130 and KDELr. Images were acquired on a Zeiss 880 confocal microscope, and z-stacks collapsed to draw regions outlining Golgi staining. Images were analyzed using FIJI (ImageJ) software. To measure the fraction of the nucleus circumference that was covered by the Golgi, the length of the nuclear perimeter adjacent to the Golgi was measured, as was the full nucleus circumference. The ratio of the length of the Golgi along the nucleus, divided by the length (circumference) of the nucleus, was quantified and plotted. To measure compaction of the Golgi, the area and perimeter of the Golgi were measured based on GM130 staining, and the circularity formula (C = (4*π*Area) / (Perimeter^2)) was applied.

To quantify ERGIC53 surfaces, confocal images of MEFs stained using GM130 and ERGIC53 antibodies were segmented in Imaris after local background subtraction. Information was extracted so as to obtain the distance of each ERGIC53 surface from a Golgi surface. The ERGIC53 surfaces that were not touching any of the Golgi surfaces were considered as independent ERGIC53 surfaces. The number of such independent surfaces was measured in each cell. Data collected on a cell-by-cell basis were plotted and analyzed.

To monitor mean ERGIC53 intensity at the cis-Golgi, confocal stacks were collapsed. Regions of interest (ROIs) that represent the Golgi area were generated from GM130 images by using the Analyze Particles function in FIJI (ImageJ). ROIs were then overlaid on the ERGIC53 channel to compute mean ERGIC53 intensity. ROIs that covered the largest contiguous Golgi signal were used for analysis.

To quantify relative enrichment of mKate2-tagged PAUF upon pulse release, ROIs were drawn in the region corresponding to the Golgi apparatus to obtain mean signal intensity from the Golgi, which was then normalized to the surrounding signal from the ER. Care was taken to exclude the nucleus which does not contain any signal. For each cell signal enrichment at the Golgi was represented as a ratio of the signal at the Golgi to that divided by the signal in the surrounding ER region.

To analyze the distributions of ERES in relation to the cis-Golgi, cells fluorescence micrographs were captured using a SoRa CSU-W1 with an inverted microscope (Nikon Ti2-E) and ORCA-FusionBT back-thinned camera (Hamamatsu). Z-stacks with 0.2 μm increments, covering 9 μm from the bottom of the cell were obtained. Three emission channels were used: one to image anti-Sec31A antibody signals, one for anti-GM130, and one for DAPI signals. For analysis of GM130 and Sec31 immunofluorescence signals using ImageJ and FIJI, immunofluorescence signal was defined as top 2.5% intensity of whole confocal image fluorescence. The single threshold value among every confocal image under analysis was calculated and used for the subsequent analysis. Plugins called "3D OC Options" and "3D Objects Counter" were used to calculate the puncta counts and voxels of GM130 and Sec31A per cell. To calculate Sec31A signals associated with GM130, double-channel-positive voxels were selected and used to calculate the puncta count and volume; puncta bigger than 10 voxels (above minimum confocal resolution 200 nm × 200 nm × 200 nm) were defined as true double-positive puncta. The count and volume of double-positive puncta were used to calculate Sec31A count (% of total Sec16A count) in GM130 and Sec31A volume in GM130 (% of total Sec31 volume).

All 4Pi-SMS images were rendered using Point Splatting mode (10 nm particle size) with Vutara SRX 7.0.06 software (Bruker, Germany). Line-scan profiles were generated by custom code in Fiji-ImageJ and plotted and curve-fitted by custom code in Python 3, as described[61] and as in Supplementary information.

To monitor the average distances between per cell between the cis and the trans Golgi, 3D isosurfaces for cis and trans were generated separately in PYME-Visualize using octree[88]. Cis (GM130) isosurfaces were further processed with shrink wrapping algorithm[89] so they better fit the GM130 point clouds. The parameters of the isosurface creation and shrinkwrapping algorithms were adjusted such that the 3D isosurfaces enclose the imaged localizations closely. The 3D isosurface meshes were then processed using MeshLab (www.meshlab.net)[90]. Faces were inverted and small disconnected components were removed.

After generating meshes, we discarded G97 + p230 points that were not inside the trans mesh and used the points in the trans mesh to calculate their shortest distances to the cis meshes. We used ray casting to determine if a point is inside a mesh, and KDTree to compute the shortest distance from each point to the mesh surface. The distances to cis mesh were used to make histogram and violin plot for each cell. Data > 500 nm were cut off because they were likely not results from the same Golgi stack. Data processing and plots were done with Python 3.10 using trimesh (v4.5.3), pandas (v1.5.1), NumPy (v1.23.4), matplotlib (v3.6.2) and SciPy (v1.9.3) packages.

To quantify the differences in localization of GFP-CFTR in WT and TUG KO MEFs, Pearson's correlations on images were analyzed and quantified using the ImageJ colocalization module.

To quantify signals from images of western blots and in gel fluorescence, images were subjected to background subtraction. ROIs were drawn around the bands of interest and integrated signal intensity was quantified using ImageJ.

Figures were prepared using FIJI (ImageJ2) v. 2.14.0/1.54 f, Adobe Photoshop 25.12.0, and Adobe Illustrator 28.7.1. GraphPad Prism v. 10.4.2 was used for plots and statistical analyses.

## Protein expression, purification of His-tagged proteins
Plasmid coding 6xHis- and mCherry-tagged TUG and the N-terminal deletion mutant and 6xHis- and mNG-tagged TUG were transformed into *E. coli* BL21 cells. Bacteria were grown in LB medium with 100 µg/ml ampicillin first in 10 ml overnight at 37 °C and then was used to inoculate 1 L LB medium with 100 µg/ml ampicillin for 4 h at 37 °C. Cells were then shifted to 16 °C for 10 min and were induced by the addition of 0.1 mM IPTG and incubated overnight at 16 °C. The culture was harvested and resuspended in protein purification buffer (25 mM Tris–HCl (pH 7.4), 500 mM NaCl, 5% glycerol, and 1 mM DTT) containing cOmplete protease inhibitor cocktail (Roche), then lysed using a high-pressure homogenizer. The crude lysate was centrifuged at 30,000 rpm for 45 min at 4 °C in a Ti-70 rotor. Supernatant was applied to a prepacked column with Ni-NTA resin (Qiagen), which was equilibrated in protein purification buffer also containing 5 mM imidazole. Protein was eluted after washing the column with buffer containing up to 20 mM imidazole, then eluted with 250 mM imidazole. Eluted fractions were pooled together and concentrated before further purifying using gel filtration using a Superdex 200 size-exclusion column (GE Healthcare) and equilibrated in the buffer with a composition 25 mM Tris–HCl (pH 7.4), 500 mM NaCl, 5% glycerol, and 1 mM DTT. Protein was stored at −80 °C after flash freezing in liquid nitrogen.

## Condensate formation assays
To monitor condensate formation, proteins were thawed, and buffer exchanged to assay buffer (25 mM Tris, 125 mM NaCl and 1 mM DTT) using Amicon ultra centrifugal filters with a 30 kDa cutoff. Protein was diluted to the final desired concentration in the assay buffer, either in the presence or absence of Ficoll 400 (Sigma; F2637) and incubated for 5 min in PCR tubes. The solution was then plated on to glass bottom imaging dishes from Cellvis (D35-14-1.5-N) and condensates were imaged on Zeiss 880 using a 63x/1.4 oil objective. To monitor the fusion of two or more condensates, a continuous time-lapse image series was recorded.

## FRAP measurements on condensates
To monitor the fluidity of the condensates, we measured the FRAP recovery after photobleaching a small region within the condensate. Approximately, 40 µM protein was incubated in the presence of 10% Ficoll 400 and incubated for 3 min before plating the solution of glass bottom imaging dishes from Cellvis. Condensates which had settled down on the bottom-most plane were imaged on Zeiss 880 using a 63x/1.4 oil objective. To photobleach a small region, an ROI was selected within the condensate which was bleached using the 561 nm laser operated at 100% power. A continuous stream of images was then acquired using the 561 nm laser at 0.5% laser power to monitor the recovery within the bleached spot. To obtain the FRAP recovery curve, the intensity within the bleached region was plotted as a function of time. Bleach correction was applied by normalizing the intensity within the bleached region with the same ROI being placed in an unbleached region from the same time series.

## FRAP measurements on mitochondrially tethered TUG
To monitor the fluidity of the TUG assemblies when tethered to the mitochondrial, we transfected TUG KO HeLa cells with the full-length TUG protein containing the mitochondrial targeting sequence and appended with a mCherry tag. Forty-eight hours transfection, cells were incubated in complete DMEM without phenol red and imaged on Zeiss 880 using a 63x/1.4 oil objective. To photobleach a small region, an ROI was selected within the condensate which was bleached using the 561 nm laser operated at 100% power. A continuous stream of images was then acquired using the 561 nm laser at 0.6% laser power to monitor the recovery within the bleached spot. Data analysis was carried out as described above for in vitro experiments.

## Reporting summary
Further information on research design is available in the Nature Portfolio Reporting Summary linked to this article.

## Data availability
All methods and data supporting the findings are available within the manuscript or supplementary information. The mass spectrometry proteomics data have been deposited to the ProteomeXchange Consortium via the PRIDE partner repository (https://www.ebi.ac.uk/pride/) with the dataset identifier PXD064240. Source data are provided with this paper.

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

## Acknowledgements

The authors thank James Rothman, Thomas Melia, Devin Fuller, Fred Gorelick, Felix Rivera-Molina, Joerg Bewersdorf, Julia von Blume, Yumei Wu, Dmytro Yushchenko, Aaron Gitler, David Stephens, William Guggino, Ke Xu, Chris Burd, Yuichi Wakana, Longhui Zheng and Dazhi Li for advice, reagents and assistance. This work used Core facilities of the Yale Diabetes Research Center (DRC, NIH P30 DK045735), services of the Yale Center for Cellular and Molecular Imaging confocal and electron microscopy facilities, Yale CINEMA imaging facility and the Keck Biotechnology Resource Center at Yale University. We would like to thank the proteomics platform (ProteoSeine), Institut Jacques Monod, Paris, France, for assistance with mass spectrometry. This work was supported by NIH grants R01 DK129466, by a Yale POINTS Pilot award and by a Pilot award from the Blavatnik Family Foundation (to J.S.B.). A.P. was supported by a Pilot and Feasibility grant from the Yale Diabetes Research Center (P30 DK045735). D.T. acknowledges support from NIH grants R01 GM151829 and R01 GM134148. I.R. acknowledges the support of Fondation pour la Recherche Médicale (AJE202210016216), Agence Nationale de la Recherche (MatSec) and the Fondation ARC.

## Author contributions

A.P. and J.S.B. conceptualized the project and designed the experiments. A.P., H.T., Z.X., M.S., Y.K., O.J.-Z., A.R.A.-R., M.V., Y.Y., X.L., D.T., I.R. and J.S.B. performed experiments and analyzed data. J.S.B. supervised the overall project and A.P. and J.S.B. wrote the manuscript with input from all authors.

## Competing interests

The authors declare no competing interests.
