## [Transparent Peer Review file · Nature Communications]

TUG protein acts through a disordered region to organize the early secretory pathway

Corresponding Author: Dr Jonathan Bogan

Version 0:

Reviewer comments:

Reviewer #1

(Remarks to the Author)

The manuscript by Parchure et al expands previous work by this Bogan lab on the role of TUG in membrane trafficking. They propose that TUG is a regulator of the spatial distribution of ERGIC membranes and thereby acts as a regulator of conventional and unconventional protein trafficking. Overall, the manuscript is well written and the story is straight-forward. The topic is interesting and of great interest to the trafficking community. However, in some instances, I think that the data do not support the conclusion and leave too much room for alternative interpretations and models. I have added below major and minor points that need to be addressed before further considering this manuscript for publication.

Major points:

1- The way I understood the authors, they propose that ERGIC-53 localizes only localizes in the Golgi region when TUG is absent. This contradicts at least 30 papers that I am aware of. In our hands, as in the hands of many others, ERGIC-53 is always in the center of the cell as well as in peripheral puncta. Here just some examples (the list could be extended much more, and I just picked randomly papers from different labs):

PMID: 33763635

PMID: 24481816

PMID: 24237698

PMID: 18287528

PMID: 15632110

PMID: 19632984

I think that the absence of TUG results in a stronger clustering of ERGIC-53 in the Golgi region, and the overexpression of TUG abolishes this localization. However, ERGIC-53 is always clustered. I think the authors should change the wording to reflect this.

2- What about the clustering of ERES? This should be demonstrated using different markers such as SEC16A and TANGO1. The authors should show that overexpression of TUG disperses the ERGIC, without affecting ERES. If ERES follow the same pattern of the ERGIC, then it would affect the interpretation of subsequent experiments. If ERES cluster less in the Golgi region, then it would explain the slower trafficking in TUG (over)expressing cells. Conversely, stronger clustering of ERES in TUG-KO cells would explain faster trafficking, because ERES are closer to the Golgi.

3- The authors claim that TUG-mCherry was excluded from the perinuclear ERGIC. This is not really shown. What the data show is the overexpression of TUG, results in a reduction (or loss??) of juxtannuclear ERGIC-53 localization.

4- Figure S1: The colocalization between TUG and Stx12 is not convincing. There is only very little overlap, and I am not sure whether it is specific. The vesicles labelled with Stx12, are mostly negative for TUG.

5- I recommend confirming the ER-Golgi trafficking result with a RUSH assay and ideally with an endogenous cargo if possible.

6- The autophagy experiments are a bit underdeveloped. I think it is important to work with endogenous autophagy readouts. The authors could check whether the initiation of autophagy is altered by looking at LC3 lipidation. They could also monitor the levels of endogenous p62, to determine the autophagic flux. Many other possibilities exist and I think that more validation

is needed.

7- The results for the CFT trafficking need more controls. Another explanation for this result might be that TUG controls exit of CFTR from the Golgi. To demonstrate the role of TUG in this Golgi-bypass pathway beyond doubt, the authors should follow the trafficking of CFTR in living cells (e.g. by RUSH) and determine whether it follows a different route in control and TUG-KO cells.

Minor points:

1- In the Introduction, the authors write that the "ERGIC was originally considered as responsible for the long-range traffic of cargoes from peripheral ERES to the Golgi". I think that this is not correct. This was actually the view of the Saraste lab, but the majority of the community debated two other questions on the nature of the ERGIC: some considered the ERGIC a trafficking intermediate (for any kind of ER-to-Golgi traffic, i.e., independently of whether it is peripheral or central ERES). Others, considered the ERGIC a stable compartment, acting as a sorting station in bidirectional ER-to-Golgi traffic (again for traffic from peripheral and central ERES). I think that this question has so far not been completely resolved. The authors may refer to these two opposing views of the ERGIC and what it was considered to do. However, the issue with long-range transport from peripheral ERES is not supported by the majority of data in the literature. Plenty of ERGIC clusters can also be observed in the Golgi region.

2- How can ERGIC-53 be in the nucleus? Are the authors proposing the existence of ERGIC-53 containing vesicles in the nucleus?

3- Figure 1d: I would like to encourage the authors to add some examples for this to the supplement

4- Is the central IDR of TUG really specifically targeted to the nucleus, or does the protein simply go to the nucleus because of its size?

Reviewer #2

(Remarks to the Author)

In the article titled "TUG protein acts through a disordered region to organize the early secretory pathway," Parchure and colleagues investigate the role of TUG in organizing the ER-Golgi Intermediate Compartment (ERGIC) and its function in cells physiology. The authors show in cells that TUG localizes to the ERGIC, and its knockout enhances anterograde secretory flux, results in redistributed ERGIC markers, and alters Golgi morphology. Targeting of TUG to the mitochondria similarly alters mitochondrial morphology, support the role of TUG in maintaining membrane-bound organelle morphology. The authors use in vitro reconstitution to show that TUG can undergo phase separation and use this data to support their suggestions that TUG puncta are in fact condensates. Functional, TUG deletion dysregulates autophagy and collagen secretion and results in altered CFTR cellular distribution.

This study provides an insightful look into the functional role of TUG in cells and provides evidence that phase separation may play a role in maintaining ERGIC organization and function. The introduction and discussion are clear, and conclusions well supported by the data. There are a few concerns highlighted below that may help strengthen the manuscript. If these can be addressed, this reviewer will support the publication of this study in Nature Communications.

Major Concerns:

1) In figures 2e and f, the authors use line scans were shown of select regions for each WT and KO image. What is the average distance between GM130 and Golgin97 across the entire image? There appears in WT and KO that the distances vary, so it would be interesting to know the averages.

2) In in vitro reconstitution assays, why was mCherry used, as it is known to oligomerize and, thus, contribute to phase separation by increasing valency. His-tagged TUG should have either been fused to mNeonGreen, which has a much higher propensity to remain monomeric or be labeled using an organic dye to avoid confounding effects of mCherry self-association. TUG condensation should be tested using either of these labeling techniques to ensure that mCherry fusion is not generating artifactual condensation. While following cellular experiments are evidence that mCherry is not artifactually contributing to TUG phase separation, cellular condensation could be caused by binding partners in the complex cellular environment rather than TUG itself.

3) Regarding the text in lines 224-228, "The spatial distribution of the FUS protein N-terminal 224 IDR, which may also form biomolecular condensates, was distinct from that of TUG-IDR1 (Fig. 3e). Together with data showing that TUG IDR2 is present in a diffuse pattern (Fig. 3c, above), we conclude that the IDR1 structures are specific and not due to a general feature of expressing intrinsically disordered protein domains in cells," perhaps the authors could cite recent molecular grammar studies and offer a brief explanation for why TUG is unique from other condensates. As far as this reviewer knows, there are no studies claiming a general IDR-driven phase separation. Unique condensates should perform unique functions, much like their membrane-bound counterparts. The text should reflect this reality. Please see PMIDs 29961577, 36603581, 39232584 for more information.

Minor Concerns:

1) In all figures, please use magenta and green instead of red and green to improve access for color blind readers.

- 2) Regarding the statement in lines 222-224, "Due to submicron size of the puncta, it was not possible to photobleach only a small portion within an individual spot and to monitor fluorescence recovery within the condensate," if the condensates are liquid-like, there should be some recovery of the entire condensate, even though they are sub-micron. Photobleaching of condensates and total recovery should be reported even though bleaching of a small region within the condensate is not feasible. These results will speak to the dynamics of the TUG molecules within punctate structures.
- 3) The use of the word "signals" in lines 241 and 243 are a bit confusing. It may be better to use "sequences" instead.
- 4) A schematic model in Fig. 4 or the supplement would be helpful to visualize the model the authors propose based on their data summarized in lines 259-261.
- 5) In line 265, "is typically mediated" would be better stated as "can be mediated by." Domain-motif interactions play a huge role in condensate formation and shouldn't be so easily discounted.
- 6) In lines 283 and 284 describing Fig. 4i, j, the authors state "Compared to the wild-type protein under same conditions, the mutant formed smaller droplets, covering less area in the imaging field of view (Fig. 4i, j)." These data could be the result of oligomerization vs. larger-scale phase separation as posited by the authors. However, they could also be caused by a decrease in the valency of TUG, leading to less condensation, similar to what has been previously observed in phase separating systems, i.e., PMID 27056844.
- 7) As a summary of their work, it would be helpful for the authors to generate a model of TUG function and how examples of CFTR and GLUT4 fits with their model either in Figure 5 or in the supplement. Preferably in Figure 5.

Reviewer #3

(Remarks to the Author)

In this manuscript, Parchure et. al. aim to reveal a more general function for TUG. While TUG is known to play a critical role in GLUT4 trafficking in adipocytes and muscle cells, its functions in other cell types remain unclear. The authors used fluorescence microscopy in combination with transmission electron microscopy (TEM) and tomography to assess the spatial organization of early secretory compartments and the Golgi in wildtype (WT) and TUG knockout (TUG-KO) cells. Additionally, they performed structure-function analyses, revealing that the TUG intrinsically disordered region (IDR) is sufficient for ERGIC localization, and that the N-terminal region of TUG is responsible for TUG oligomerization in trans. Finally, they demonstrated that TUG regulates autophagic flux, as well as collagen and CFTR trafficking. While this manuscript attempts to provide a general overview of TUG's roles in the secretory pathway, the data are primarily descriptive and lack the depth needed for a comprehensive understanding. The logical flow between figures is unclear, and as the manuscript currently stands, I would not recommend it for publication in Nature Communications.

Major Concerns:

1. Fluorescence Microscopy Data: The fluorescence microscopy data are largely descriptive and often lack proper reference markers. A thorough investigation of TUG-mCherry localization in relation to various secretory pathway markers, such as ERESs and Golgi, should be performed in both WT and TUG-KO cells. Establishing a quantitative baseline for these markers early in the manuscript would provide stronger evidence for the story being presented.
2. Inconsistent Golgi Fragmentation: In the literature, most cell types in resting state have a central, compact Golgi. In Figure 1c, the Golgi in WT MEFs appears fragmented, yet in Figure 1f, the Golgi in WT MEFs is shown as more compact. This discrepancy between the images needs to be addressed for consistency. I recommend presenting individual images for each channel, in addition to the merged image, to clarify this issue.
3. Secretory Flux: The Ss-mKate-FM4-PAUF experiment is an interesting approach to examine secretory flux. However, the lack of reference markers (for the ER, ERES, ERGIC, and Golgi) makes the experiment less convincing. Moreover, without the inclusion of other cargo systems or an orthogonal approach, the data supporting the claim that "TUG acts as a brake on flux....." is a bit thin. This data seems contradictory to the findings in Figures 5e-h, which show defective collagen and CFTR secretion. Although the authors discuss these issues and propose several potential explanations in the discussion, the effects of TUG KO on collagen and CFTR secretion appear to be indirect. I am unsure whether these data contribute to understanding the mechanisms by which TUG regulates protein secretion or if they raise more questions than answers.

Minor Concerns:

1. Structural Analysis of TUG: The structural analysis of TUG is well-conducted. However, I recommend including some level of quantification and incorporating reference markers for secretory compartments to strengthen the results.
2. Phase-Boundary Analysis: Incorporating phase-boundary analysis would enhance the data presented in Figures 4i and 4j.

Version 1:

Reviewer comments:

Reviewer #1

(Remarks to the Author)

The authors have responded to all points that I initially raised. Although it is a pity that some of the experiments I asked for could not be done (e.g. the RUSH assay or the trafficking of an endogenous cargo). I nevertheless thank the authors for trying solve these question. Overall, the manuscript improved and the critical points have been addressed to my satisfaction. I therefore have no further points and think this work should be published now.

Reviewer #2

(Remarks to the Author)

In the resubmitted version of their manuscript, the authors have addressed the concerns of this reviewer. The attention to detail in the text and responses is appreciated. The additional data the authors provided comparing mCherry vs. mNeonGreen is convincing and shows that the fluorophore has negligible effect on the phase separation of TUG. The FRAP experiments also provide additional evidence that TUG is forming condensates, as the fluorescence recovery is consistent with condensed material on intracellular membranes. The additional graphic models in the supplementary material are also helpful for visualizing the pathway. Finally, the method for calculating distances in supplemental figure 4 is an excellent addition and clearly explains the methodology while providing a road map for other researchers to apply this method to their research. With the additional work on this manuscript, this reviewer now supports publication in Nature Communications.

Reviewer #3

(Remarks to the Author)

Overall, the revised manuscript has significantly improved. The authors have addressed most of my concerns properly. The only concern remains related to the secretory flux experiment. The authors have invested a significant amount of time and efforts to develop assays for a new RUSH reporter in MEFs. I understand that sometimes the RUSH system simply does not work in certain cells, as we have encountered similar situation in our own work. That said, there are many other existing RUSH reporters to serve this purpose. Additionally, the authors also have a HeLa cell system for addressing the secretory flux upon TUG depletion. These experiments are crucial to substantiate the claim that TUG KO MEFs have greater ER-Golgi cargo flux (line 168-171).

In its current form, the data seem to point toward a cargo-specific effect: collagen and CFTR secretion were significantly reduced, while small and soluble cargo proteins appeared to be unaffected (Figure 6). If TUG truly acts as a general “brake” on early secretory pathway trafficking, one would expect a broader impact across cargo types. Therefore, the findings are more consistent with a selective role of TUG in regulating certain cargoes, rather than a general inhibition of ER–Golgi transport.

Responses to Reviewer Comments

We appreciate the three reviewers' thoughtful analysis and comments. All comments are reproduced below with our point-by-point responses in blue underneath each comment.

Reviewer #1 (Remarks to the Author):

The manuscript by Parchure et al expands previous work by this Bogan lab on the role of TUG in membrane trafficking. They propose that TUG is a regulator of the spatial distribution of ERGIC membranes and thereby acts as a regulator of conventional and unconventional protein trafficking. Overall, the manuscript is well written and the story is straight-forward. The topic is interesting and of great interest to the trafficking community. However, in some instances, I think that the data do not support the conclusion and leave too much room for alternative interpretations and models. I have added below major and minor points that need to be addressed before further considering this manuscript for publication.

We thank the reviewer for an overall positive assessment of the manuscript. We have addressed the points that raised by the reviewer, as described below. We think this has strengthened the manuscript, and we are grateful for the thoughtful critique. We hope that the reviewer will support publication of the revised manuscript.

Major points:

1- The way I understood the authors, they propose that ERGIC-53 localizes only localizes in the Golgi region when TUG is absent. This contradicts at least 30 papers that I am aware of. In our hands, as in the hands of many others, ERGIC-53 is always in the center of the cell as well as in peripheral puncta. Here just some examples (the list could be extended much more, and I just picked randomly papers from different labs): PMID: 33763635, PMID: 24481816, PMID: 24237698, PMID: 18287528, PMID: 15632110, PMID: 19632984. I think that the absence of TUG results in a stronger clustering of ERGIC-53 in the Golgi region, and the overexpression of TUG abolishes this localization. However, ERGIC-53 is always clustered. I think the authors should change the wording to reflect this.

We apologize for the confusion, and we agree with the reviewer that as written it was confounding. Our intention was to emphasize that the Em-ERGIC53 construct shows the expected distribution, extending to the ERES and cis-Golgi, when it is expressed on its own. Nevertheless, we agree that ERGIC-53 is clustered, and it remains so in the absence of TUG and with TUG overexpression. We now clarify that when ERGIC-53 is co-expressed with TUG-mCherry, there is colocalization of these proteins in clustered cytosolic punctate structures. This has been stated more clearly in the revised manuscript.

2- What about the clustering of ERES? This should be demonstrated using different markers such as SEC16A and TANGO1. The authors should show that overexpression of TUG disperses the ERGIC, without affecting ERES. If ERES follow the same pattern of the ERGIC, then it would affect the interpretation of subsequent experiments. If ERES cluster less in the Golgi region, then it would explain the slower trafficking in TUG (over)expressing cells. Conversely, stronger clustering of ERES in TUG-KO cells would explain faster trafficking, because ERES are closer to the Golgi.

We thank the reviewer for this suggested experiment, which we were able to undertake after receiving a custom antibody to label Sec31A in MEFs (from Dr. Fred Gorelick, Yale). This was

important, as our pilot studies showed that commercial Sec31 antibodies and other antibodies to ER exit site markers did not work well in MEFs for immunofluorescence. As suggested by the reviewer, we fixed cells that were either i) wildtype (WT), ii) TUG KO and iii) WT MEFs with TUG overexpression and labeled them using antibodies to Sec31A and GM130. We imaged these cells by confocal microscopy, and segmented the Golgi and the ERES spots using a custom Fiji code. This enabled us to count the number of Sec31A puncta that are touching the Golgi, versus the total number of Sec31A clusters in a cell. The images and the quantification are presented in Supplementary Fig. 3. We did not observe any significant changes in the absolute number or percentage of ERES touching the cis-Golgi in TUG KO cells or in TUG overexpressing cells, compared to WT controls. Due to Golgi compaction in the TUG KO cells (shown in the original version of the manuscript and now in Fig. 3a-c), there may be an increase in the density of ERES in the vicinity of the Golgi. Yet, importantly, the number of the ERES in the vicinity of the Golgi was unchanged, and thus the accelerated flux we observed cannot be attributed to any large change in ERES distribution. This point is now stated in the description of the results.

3- The authors claim that TUG-mCherry was excluded from the perinuclear ERGIC. This is not really shown. What the data show is the overexpression of TUG, results in a reduction (or loss??) of juxtannuclear ERGIC-53 localization.

We had originally included data in a supplementary figure showing that overexpressed TUG does not localize to the Golgi apparatus, as marked using GMAP210-mNG. In the revised manuscript, we have now added a high-resolution image (Fig. 1c) to show that TUG-mCherry does not colocalize with co-expressed GMAP210-mNG in TUG KO HeLa cells, and we also include data showing that this result is also obtained in WT HeLa cells (Supplementary Fig. 1e). We have also changed the wording to make clear that overexpressed TUG-mCherry is excluded from the Golgi region. This is different from the distribution of Em-ERGIC53, which overlaps with GM130 (Supplementary Fig. 1a).

4- Figure S1: The colocalization between TUG and Stx12 is not convincing. There is only very little overlap, and I am not sure whether it is specific. The vesicles labelled with Stx12, are mostly negative for TUG.

We have added a better representative image that shows overlap of TUG-mCherry and GFP-Stx12 structures and highlights some of these structures (Fig. 1d). We agree that Stx12 is also present in other structures, not overlapping with TUG-mCherry, which we indicate in the text. Given that TUG-mCherry marks ERGIC53 - positive membranes that are distinct from the ERES (Fig. 1b), having an additional marker of structures labeled by TUG-mCherry provides better characterization of the compartment, strengthening the overall result.

Presently, we do not know what interaction partner localizes TUG to ERGIC53 -positive membranes. In experiments that are in progress, we see that *in vitro*, TUG-mCherry can bind to phospholipids, suggesting it may interact directly with membranes. We tested this by doing floatation assays. We incubated purified, mCherry-tagged TUG with liposomes having a composition resembling ER membranes (which are very similar to ERGIC membranes, according to the literature) and saw TUG floating up (Reviewer Fig. 1). We also have

done an interactome analysis, using tandem mass spectrometry, and in preliminary results this identified candidates that may help localize TUG to the ERGIC. We are pursuing these candidates further. Full exploration of these preliminary data will require substantial effort, and is beyond the scope of the current manuscript (and of the four-month timeframe for revision specified by the editor).

5- I recommend confirming the ER-Golgi trafficking result with a RUSH assay and ideally with an endogenous cargo if possible.

We attempted to confirm the results of experiments in which we monitored the kinetics of transport from the ER to the cis-Golgi using RUSH assay. For this purpose, we amplified the LyzC-SBP-eGFP RUSH reporter constructs from the von Blume laboratory at Yale University (used in PMIDs: 32422653, 32479594, and others) and cloned them under the retroviral promoter in the pCX4 vector. This is the same vector we used to express the FM4 domain - containing reporter construct, which enabled us to successfully observe differences in transport kinetics in TUG KO and wildtype MEFs. As expected, without addition of biotin, the LyzC reporter was retained in the ER. However, upon addition of biotin, we observed marked cell-to-cell heterogeneity in the localization of this construct (Reviewer Fig. 2, below). This heterogeneity persisted, even 30 minutes after biotin addition. As illustrated in Reviewer Fig. 2, whereby in some cells the reporter is still retained in the ER while in others it is present at the Golgi. We have therefore limited confidence in using this system in MEFs for robust kinetic analysis, especially when we have presented data from a similar ER-release approach that experimentally works much more robustly. We note that the cell-to-cell variability that we observed is not due to differences in expression, as the expression levels are low to begin with, due to the use of a relatively weak retroviral promoter. While in theory other reporters or vectors might work better, technical development is beyond the scope of this manuscript, particularly given the 4-month time frame for revisions.

Reviewer Fig 2. Panel of images showing cell to cell variability in cargo localization in attempted RUSH experiments. Cells were fixed at time points 20 or 30 minutes after addition of biotin, as indicated, then fixed and stained using antibodies to GM130 and the GFP booster to amplify the signal from the reporter. While in some cells, cargo is at the Golgi, in others it is still in the ER. Without addition of biotin, cargo is in the ER.

6- The autophagy experiments are a bit underdeveloped. I think it is important to work with endogenous autophagy readouts. The authors could check whether the initiation of autophagy is altered by looking at LC3 lipidation. They could also monitor the levels of endogenous p62, to determine the autophagic flux. Many other possibilities exist and I think that more validation is needed.

We agree with the reviewer that the assay we used does not distinguish whether the reduction in autophagy flux in TUG KO cells is due to a defect in autophagosome biogenesis, or due to a defect in fusion of autophagosomes with lysosomes. Each of these possibilities could cause reduced autophagic flux, as we observed in TUG KO cells. We have added new data in the revised manuscript to support the idea that there is impaired autophagosome biogenesis, as noted below.

We would like to point out some strengths of the LC3 reporter assay we used in the original version of our manuscript. Although this assay relies on ectopic expression of a reporter protein, the flux measurements are obtained as a ratio of fluorescence intensities and are highly sensitive. In addition, since we only monitor the reporter after a brief (20 minute) pulse label, results are not confounded by transcriptional upregulation of p62 during starvation (PMID: 24394643) and the assay does not require addition of Bafilomycin A1.

To address the question of why autophagy flux is reduced in TUG KO MEFs, we starved WT and KO cells in presence of Bafilomycin A1. In WT cells, this results in accumulation of autophagosomes labeled using an antibody to detect endogenous LC3B, since autophagosome fusion with lysosomes is blocked by Bafilomycin A1. We hypothesized that, in TUG KO cells, a defect in autophagosome biogenesis would result in reduced accumulation of LC3B -positive autophagosomes in this assay. This is exactly what we observed, and the images and quantification are included in Figs. 6d and 6e of the revised manuscript. Moreover, in images obtained under basal conditions, shown Supplementary Fig. 8b, very few autophagosomes (LC3B positive structures) are observed. Together with other results, these data support the idea that disruption of the ERGIC in TUG KO cells results in impaired autophagosome biogenesis and, consequently, reduced autophagy flux.

7- The results for the CFTR trafficking need more controls. Another explanation for this result might be that TUG controls exit of CFTR from the Golgi. To demonstrate the role of TUG in this Golgi-bypass pathway beyond doubt, the authors should follow the trafficking of CFTR in living cells (e.g. by RUSH) and determine whether it follows a different route in control and TUG-KO cells.

We agree with the reviewer that the altered localization of CFTR we observed could be due to a defect in Golgi exit. It is very challenging to prove beyond doubt that this altered CFTR targeting results from effects on a Golgi bypass mechanism, using the experiments suggested by the reviewer, for multiple reasons. The initial experiment itself was challenging, considering the GFP-CFTR construct is quite large, which makes it difficult to package in retroviruses. We therefore had to rely on transfection, which has very low efficiency in MEFs. Despite these limitations, we were able to obtain data from a sufficient number of cells, from multiple experiments, to present the data in **Figs. 6h and 6i**. Aside, the addition of more DNA elements to make a RUSH construct would further compound the issues with transfection and, as noted above, already with a smaller reporter protein we observed highly variable results by RUSH.

Accordingly, we have modified the text in the revised manuscript to indicate that we observe defects in targeting of a cargo, CFTR, that was previously described to traffic, at least in part, using a Golgi bypass pathway. Of note, we also have now analyzed the secretion of soluble GFP and did not find any difference in TUG KO MEFs, compared to WT control cells. In fact, in 2 out of 3 experiments, we observed a slight increase in GFP secretion from TUG KO cells. We think that the specificity of transport defects associated with particular cargoes is an important point of our manuscript. We hope the reviewer will agree, but we are willing to remove the CFTR data from the manuscript if the reviewer feels strongly on this point.

Minor points:

1- In the Introduction, the authors write that the “ERGIC was originally considered as responsible for the long-range traffic of cargoes from peripheral ERES to the Golgi”. I think that this is not correct. This was actually the view of the Saraste lab, but the majority of the community debated two other questions on the nature of the ERGIC: some considered the ERGIC a trafficking intermediate (for any kind of ER-to-Golgi traffic, i.e., independently of whether it is peripheral or central ERES). Others, considered the ERGIC a stable compartment, acting as a sorting station in bidirectional ER-to-Golgi traffic (again for traffic from peripheral and central ERES). I think that this question has so far not been completely resolved. The authors may refer to these two opposing views of the ERGIC and what it was considered to do. However, the issue with long-range transport from peripheral ERES is not supported by the majority of data in the literature. Plenty of ERGIC clusters can also be observed in the Golgi region.

Thank you for this comment. The relevant section of the introduction has been revised according to the reviewer’s suggestion.

2- How can ERGIC-53 be in the nucleus? Are the authors proposing the existence of ERGIC-53 containing vesicles in the nucleus?

We do not propose that ERGIC53 is in the nucleus. We typically present images that are a single confocal plane.

These planes are chosen so that punctate structures are in focus, yet there may still be out-of-focus light from large structures such as the nucleus that are present. To illustrate this point, images in Reviewer Fig. 3 are shown. In a Z-stack, an IDR-mCherry positive puncta appears to be inside the nucleus in XY plane, but is clearly above the nucleus when the entire stack is shown using orthogonal views. When the nucleus is well in focus, in subsequent slices, the puncta are not in focus.

3- Figure 1d: I would like to encourage the authors to add some examples for this to the supplement

Done. Please see Supplementary Fig. 2b. Do note that the surface is at the center of the magenta circles (ERGIC53) and pooled over 1 micron slice.

4- Is the central IDR of TUG really specifically targeted to the nucleus, or does the protein simply go to the nucleus because of its size?

The central IDR indeed harbors an NLS. We have made a construct in which we deleted the NLS (residues 294-302 of the intact mouse TUG protein) and this construct is not enriched in the nucleus, compared to the wildtype central IDR, but rather only enters passively. This is shown in Reviewer Fig. 4.

Reviewer #2 (Remarks to the Author):

In the article titled "TUG protein acts through a disordered region to organize the early secretory pathway," Parchure and colleagues investigate the role of TUG in organizing the ER-Golgi Intermediate Compartment (ERGIC) and its function in cells physiology. The authors show in cells that TUG localizes to the ERGIC, and its knockout enhances anterograde secretory flux, results in redistributed ERGIC markers, and alters Golgi morphology. Targeting of TUG to the mitochondria similarly alters mitochondrial morphology, support the role of TUG in maintaining membrane-bound organelle morphology. The authors use in vitro reconstitution to show that TUG can undergo phase separation and use this data to support their suggestions that TUG puncta are in fact condensates. Functional, TUG deletion dysregulates autophagy and collagen secretion and results in altered CFTR cellular distribution.

This study provides an insightful look into the functional role of TUG in cells and provides evidence that phase separation may play a role in maintaining ERGIC organization and function. The introduction and discussion are clear, and conclusions well supported by the data. There are a few concerns highlighted below that may help strengthen the manuscript. If these can be addressed, this reviewer will support the publication of this study in Nature Communications.

We thank the reviewer for an overall positive assessment of the manuscript. We appreciate the constructive comments and have addressed the points, as described below. We believe this has strengthened the manuscript. We hope the reviewer will now support publication.

Major Concerns:

1) In figures 2e and f, the authors use line scans were shown of select regions for each WT and KO image. What is the average distance between GM130 and Golgin97 across the entire image? There appears in WT and KO that the distances vary, so it would be interesting to know the averages.

We agree with the reviewer that there is indeed heterogeneity in the distances between the *cis* and the *trans* Golgins. Therefore, to systematically compute the distances across the entire Golgi, as the reviewer indicated, we imaged 4 more cells wherein we labeled the *trans* cisterna using a combination of p230 and Golgin97 antibodies. We then generated isosurfaces from the point cloud data and computed the shortest distances of all the *trans* localization points to the

Reviewer Fig 4. Deletion of NLS (residues 294-302 of intact TUG) from the central IDR causes it to no longer accumulate in the nucleus. Cytosolic punctate structures are still observed.

cis isosurfaces. In the revised manuscript, violin plots are presented in Fig. 3g to show the distribution of distances from *cis*- to *trans*- cisterna in four WT and four TUG KO cells.

The methodology and example images to obtain these results are also now included as a supplementary figure (Supplementary Fig. 4). Please note that this distance measurement is different from the line-scan profile measurements shown in Fig. 3e, f, which measured the center-to-center distance. The distances across the entire Golgi are computed using the shortest paths, and so will appear smaller than the line-scan profiles. In all the TUG KO cells, median distances are greater, compared to those in WT cells; indeed, the distances are much greater in 2 out of the 4 cells examined. We note that in taking averages from whole cells, there are occasionally localized increases in *cis*- to *trans*- distances. These localized areas of increased separation can be quite marked, yet they may appear less when averages are taken over the entire Golgi stack. Nonetheless, there is an increase in the median distance in all the KO cells, compared to the WT cells. The data support the idea that the distance between *cis*- and *trans*- cisterna is increased in KO cells.

2) In *in vitro* reconstitution assays, why was mCherry used, as it is known to oligomerize and, thus, contribute to phase separation by increasing valency. His-tagged TUG should have either been fused to mNeonGreen, which has a much higher propensity to remain monomeric or be labeled using an organic dye to avoid confounding effects of mCherry self-association. TUG condensation should be tested using either of these labeling techniques to ensure that mCherry fusion is not generating artifactual condensation. While following cellular experiments are evidence that mCherry is not artifactually contributing to TUG phase separation, cellular condensation could be caused by binding partners in the complex cellular environment rather than TUG itself.

Thank you for the suggestion. We purified a recombinant TUG protein tagged with mNeonGreen (mNG) instead of mCherry, and we tested the ability of this protein to form condensates in solution. In data shown in Supplementary Fig. 5d of the revised manuscript, we show that mNG-TUG forms condensates in solution in presence of 10% Ficoll 400 under physiological salt concentration (125 mM NaCl). At high salt concentration (500 mM NaCl), the protein does not form condensates.

Of note, data presented in Figs. 5i and 5j show that an N-terminal deletion of mCherry-tagged TUG abrogates the ability of the full-length to form condensates. This observation further strengthens the point that the ability of TUG to form condensates in solution is not influenced by the fluorescent tags used.

We also now include new data supporting the idea that TUG may form condensates in cells. Although the puncta formed by TUG-mCherry or IDR1-mCherry are too small to test in FRAP experiments, we have performed FRAP on mitochondrially -targeted TUG-mCherry. As shown in Supplementary Fig. 7, there is approximately 30% recovery of fluorescence within a minute after bleaching, with a major fraction of the recovery happening in the first 5-10 seconds. These data support the idea that TUG may form condensates in cells as well as *in vitro*.

3) Regarding the text in lines 224-228, "The spatial distribution of the FUS protein N-terminal 224 IDR, which may also form biomolecular condensates, was distinct from that of TUG-IDR1 (Fig. 3e). Together with data showing that TUG IDR2 is present in a diffuse pattern (Fig. 3c, above), we conclude that the IDR1 structures are specific and not due to a general feature of expressing intrinsically disordered protein domains in cells," perhaps the authors could cite recent molecular grammar studies and offer a brief explanation for why TUG is unique from

other condensates. As far as this reviewer knows, there are no studies claiming a general IDR-driven phase separation. Unique condensates should perform unique functions, much like their membrane-bound counterparts. The text should reflect this reality. Please see PMIDs 29961577, 36603581, 39232584 for more information.

Thank you for bringing up this point. We have revised the relevant text and have cited the appropriate references. We have also included a new figure, Supplementary Figure 5h, which diagrams the positions of relevant residues in TUG IDR1. The modified text now states: “we conclude that the TUG-IDR1 structures are specifically formed upon expression of this polypeptide. This may result from multivalent, dynamic interactions involving IDR1, which is rich in charged, proline, serine, and threonine residues (Supplementary Fig. 5h). This would fit with how other intrinsically disordered regions are described to be targeted to specific condensates.”

Minor Concerns:

1) In all figures, please use magenta and green instead of red and green to improve access for color blind readers.

Done.

2) Regarding the statement in lines 222-224, “Due to submicron size of the puncta, it was not possible to photobleach only a small portion within an individual spot and to monitor fluorescence recovery within the condensate,” if the condensates are liquid-like, there should be some recovery of the entire condensate, even though they are sub-micron. Photobleaching of condensates and total recovery should be reported even though bleaching of a small region within the condensate is not feasible. These results will speak to the dynamics of the TUG molecules within punctate structures.

TUG-mCherry or the IDR1-mCherry puncta are mobile in the cells. We therefore performed point FRAP measurements on TUG-mCherry which is tethered to the mitochondria and results in clumping of the mitochondria. We do indeed observe an approximately 30% FRAP recovery, the majority of which is in the first frame after photobleaching. The results are now presented in Supplementary Figs. 7b,c.

3) The use of the word “signals” in lines 241 and 243 are a bit confusing. It may be better to use “sequences” instead.

Done.

4) A schematic model in Fig. 4 or the supplement would be helpful to visualize the model the authors propose based on their data summarized in lines 259-261.

Done. Please see Supplementary Fig. 7a.

5) In line 265, “is typically mediated” would be better stated as “can be mediated by.” Domain-motif interactions play a huge role in condensate formation and shouldn't be so easily discounted.

Thank you for pointing this out. We have changed the text as suggested.

6) In lines 283 and 284 describing Fig. 4i, j, the authors state “Compared to the wild-type protein under same conditions, the mutant formed smaller droplets, covering less area in the imaging

field of view (Fig. 4i, j).” These data could be the result of oligomerization vs. larger-scale phase separation as posited by the authors. However, they could also be caused by a decrease in the valency of TUG, leading to less condensation, similar to what has been previously observed in phase separating systems, i.e., PMID 27056844.

Thank you for this comment as well. We have modified the text, which now reads: “This observation could reflect a decrease in the valency of TUG, which might lead to less condensation⁵² [PMID 27056844]. Yet, given cellular and in vitro data supporting the idea that the N-terminal structured region of TUG oligomerizes, the simplest explanation is that this oligomerization promotes larger-scale phase separation.”

7) As a summary of their work, it would be helpful for the authors to generate a model of TUG function and how examples of CFTR and GLUT4 fits with their model either in Figure 5 or in the supplement. Preferably in Figure 5.

Thank you for the excellent suggestion. Because the model of GLUT4 trafficking is based on previous work, we felt it was better to include the overall model in the supplement (Supplementary Fig. 9). As well, we cannot entirely distinguish the mechanistic effects on CFTR trafficking (discussed in response to Reviewer 1, point 7). Accordingly, we have included every other element into our model, except for CFTR. We think it is most likely that CFTR and GLUT4 both engage shared trafficking machinery at the ERGIC. In both cases, TUG is required for the proper targeting and retention of these proteins in the early secretory pathway. Stx12 may also be involved in this mechanism, as it is required for CFTR targeting and consistently associated with GLUT4 in proteomic studies, as noted in the manuscript. The different targeting of these two proteins in TUG KO cells may result from differences in the mechanistic details of their interactions with shared trafficking machinery at the ERGIC. Understanding these mechanistic details will require future work, as noted in the discussion of the revised manuscript.

Reviewer #3 (Remarks to the Author):

In this manuscript, Parchure et. al. aim to reveal a more general function for TUG. While TUG is known to play a critical role in GLUT4 trafficking in adipocytes and muscle cells, its functions in other cell types remain unclear. The authors used fluorescence microscopy in combination with transmission electron microscopy (TEM) and tomography to assess the spatial organization of early secretory compartments and the Golgi in wildtype (WT) and TUG knockout (TUG-KO) cells. Additionally, they performed structure-function analyses, revealing that the TUG intrinsically disordered region (IDR) is sufficient for ERGIC localization, and that the N-terminal region of TUG is responsible for TUG oligomerization in trans. Finally, they demonstrated that TUG regulates autophagic flux, as well as collagen and CFTR trafficking.

While this manuscript attempts to provide a general overview of TUG's roles in the secretory pathway, the data are primarily descriptive and lack the depth needed for a comprehensive understanding. The logical flow between figures is unclear, and as the manuscript currently stands, I would not recommend it for publication in Nature Communications.

We are grateful for the reviewer's thoughtful critiques, and we have addressed the points experimentally, as well as by revision of the text. We think that the inclusion of new data and explication have substantially improved the manuscript, which identifies a new regulator of membrane trafficking at the ERGIC. The logical flow between the figures is also improved by the inclusion of new data and text in the revised manuscript. We hope the reviewer agrees and will now support its publication.

Major Concerns:

1. Fluorescence Microscopy Data: The fluorescence microscopy data are largely descriptive and often lack proper reference markers. A thorough investigation of TUG-mCherry localization in relation to various secretory pathway markers, such as ERESs and Golgi, should be performed in both WT and TUG-KO cells. Establishing a quantitative baseline for these markers early in the manuscript would provide stronger evidence for the story being presented.

We have expanded the data presented in the manuscript and now provide representative images of the localization of ectopically expressed TUG-mCherry in TUG KO HeLa cells together with Em-ERGIC53, Sec23-GFP (ERES) and GMAP210-mNG (Golgi). These images are shown in Fig. 1, and similar data obtained in WT HeLa cells is shown in Supplementary Fig. 1. As is clear from the images, there is a complete overlap between TUG-mCherry and the Em-ERGIC53 in cytosolic punctate structures, in both WT and KO cells. Furthermore, TUG-mCherry puncta are distinct from the ERES marker Sec23, and TUG-mCherry is excluded from the Golgi region. In addition, we observe a subset of GFP-Stx12 structures overlapping with TUG puncta in the cytosol (Fig. 1d). The overlap of full length TUG-mCherry with these markers is also observed upon expression of the central intrinsically disordered region, IDR1 (Figs. 4f,g). As well IDR1 does not overlap with Sec23 or GMAP210 (Supplementary Figs. 6c, d). Finally, we now show that there is partial overlap of Em-ERGIC53 with GM130 and with Sec31A in TUG KO HeLa cells, as is typical of ERGIC53 (Supplementary Figs. 1a, b). The inclusion of these additional reference markers further supports the conclusion that ectopically expressed TUG localizes to the ERGIC, and not to ERES or the cis-Golgi.

Presently, we do not know what interaction partner localizes TUG to ERGIC53 -positive membranes. In experiments that are in progress, we see that *in vitro*, TUG-mCherry can bind to phospholipids, suggesting it may interact directly with membranes. We tested this by doing floatation assays. We incubated purified, mCherry-tagged TUG with liposomes having a composition resembling ER membranes (which are very similar to ERGIC membranes, according to the literature) and saw TUG floating up (Reviewer Fig. 1). We also have done an interactome analysis, using tandem mass spectrometry, and in preliminary results this identified candidates that may help localize TUG to the ERGIC. We are pursuing these candidates further. Full exploration of these preliminary results will require substantial effort, and is beyond the scope of the current manuscript (and of the four-month timeframe for revision specified by the editor).

2. Inconsistent Golgi Fragmentation: In the literature, most cell types in resting state have a central, compact Golgi. In Figure 1c, the Golgi in WT MEFs appears fragmented, yet in Figure 1f, the Golgi in WT MEFs is shown as more compact. This discrepancy between the images needs to be addressed for consistency. I recommend presenting individual images for each channel, in addition to the merged image, to clarify this issue.

We have presented the individual images for each channel, as suggested, and the data are now shown in Fig. 2a of the revised manuscript. Of note, these immunofluorescence images show a single slice from a confocal plane and connections to structures in preceding or subsequent slices will not be apparent. The morphology of the Golgi complex in WT MEFs is actually quite

similar to what has been observed previously. As examples, prior work stained MEFs with an anti-GM130 antibody and reported images reproduced in Reviewer Fig. 5 (images from Fig. 2l of PMID 33479349 and Fig. 6b PMID 24795147). The more compacted Golgi is a clear phenotype in TUG KO MEFs and we have quantified these data in two different ways, as presented in Figs. 3b,c of the revised manuscript. As assessed both by 1) the fraction of the nucleus circumference that is covered by the Golgi, and also by 2) using a circularity formula ($C = 4 \pi \text{ area} / \text{perimeter}^2$) to quantify Golgi compaction, the TUG KO MEFs are significantly different from WT control MEFs ($P < 0.0001$ in both cases). A similar highly-compacted Golgi phenotype has been reported in cells lacking GM130 (PMID: 28028212) and in cells lacking GMAP210 (PMID: 19112494).

3. Secretory Flux: The Ss-mKate-FM4-PAUF experiment is an interesting approach to examine secretory flux. However, the lack of reference markers (for the ER, ERES, ERGIC, and Golgi) makes the experiment less convincing.

Reviewer Fig 5. Confocal microscopy images of GM130 staining in MEFs, reproduced from published reports. The images in the top panels are Fig. 2l of Dieterich et al., Scientific Reports 11:2013, 2021 (PMID: 33479349). The images in the bottom panels are from Fig. 6b of Veenendaal, et al., Biology Open 3:431, 2014 (PMID: 24795147).

We thank the reviewer for this suggestion. We have now added images of cells without the addition of the D/D solubilizer, in which the aggregated cargo is stained together with an ER-marker, PDI (Fig. 2d). In addition, at the 10-minute time-point we show co-staining of the PAUF reporter with the Golgi marker GM130. These additions make it clear that the aggregated cargo is excluded from the Golgi prior to the addition of the D/D solubilizer, and that it is subsequently concentrated at the Golgi at later time points. At 10 minutes after D/D addition, the Golgi localization is more marked in TUG KO MEFs, compared to WT control MEFs, as quantified in Fig. 2e.

Moreover, without the inclusion of other cargo systems or an orthogonal approach, the data supporting the claim that “TUG acts as a brake on flux.....” is a bit thin.

We attempted to do a RUSH assay as an additional means to validate this point, as this was also suggested by Reviewer 1. For this purpose, we amplified the LyzC-SBP-eGFP RUSH reporter constructs from the von Blume laboratory at Yale University (used in PMIDs: 32422653, 32479594, and others) and cloned them under the retroviral promoter in the pCX4 vector. This is the same vector we used to express the FM4 domain -containing reporter construct, which enabled us to successfully observe differences in transport kinetics in TUG KO and wildtype MEFs. Without the addition of biotin, the LyzC reporter was retained in the ER. However, upon

addition of biotin, we observed marked cell-to-cell heterogeneity in the localization of this construct (Reviewer Fig. 2, repeated below for convenience). This heterogeneity persisted, even 30 minutes after biotin addition. This is illustrated in the images shown in Reviewer Fig. 2, where in some cells the reporter is still retained in the ER while in others it is present at the Golgi. We have therefore no confidence in using this system in MEFs for the kind of kinetic analysis we want to do, especially when we have presented data from a similar experimental approach that works much better in our hands. We would also like to point out that the cell-to-cell variability we observed is not due to differences in expression. The expression levels are low to begin with, due to use of a relatively weaker retroviral promoter to drive expression. We think that in MEFs there would have to be an extensive characterization of an appropriate reporter and vectors to be able to obtain more consistent data. This technical development is beyond the scope of this manuscript, particularly given the 4-month time frame for revisions.

This data seems contradictory to the findings in Figures 5e-h, which show defective collagen and CFTR secretion. Although the authors discuss these issues and propose several potential explanations in the discussion, the effects of TUG KO on collagen and CFTR secretion appear to be indirect.

We have added new data, explanation, and references, to the manuscript to support the idea that effects of TUG KO are specific and thus unlikely to be indirect. In the revised manuscript, we show that secretion of soluble GFP is unaffected in TUG KO MEFs, compared to WT control cells (Figs. 6j,k). Indeed, secretion of soluble GFP was increased in KO MEFs in 2 out of 3 experiments. These new data complement our secretome analysis, which demonstrated that the reduced collagen secretion in TUG KO cells is both quite dramatic and also rather specific, since secretion of several small soluble proteins was unaffected (Fig. 6f, g).

We also have provided additional context for the alteration in CFTR targeting in TUG KO MEFs, shown in Figs. 6h,i. As noted in the response to Reviewer 2, CFTR and GLUT4 engage shared trafficking machinery at the ERGIC, but they do so in different ways. GLUT4 binds to TUG, but not to the TUG-interacting protein PIST (PMID: 22610098), whereas CFTR binds directly to the TUG-interacting protein PIST (also called CAL, PMID: 11707463, 14570915). GLUT4 goes to the cell surface in TUG KO cells, but CFTR enters the Golgi in TUG KO cells. Thus, the difference in targeting of these two proteins may result from differences in how they interact with trafficking machinery at the ERGIC. Understanding these details will require substantial future work. Yet, since both proteins interact with shared machinery, the alterations in trafficking are unlikely to be indirect effects of TUG KO. This point is now made (and references are cited) in the discussion section of the manuscript.

I am unsure whether these data contribute to understanding the mechanisms by which TUG regulates protein secretion or if they raise more questions than answers.

Our manuscript is the first work that systematically demonstrates a general role for TUG, distinct from its specialized role to control GLUT4 trafficking. Our data show that TUG acts at the ERGIC to modulate secretion; to our knowledge, it is the first-described negative regulator of cargo flux through the early secretory pathway. Its effects on autophagy, collagen secretion, and CFTR targeting are likely to be important for physiology. As with many scientific advances, the data raise many questions, and further studies will be required to understand the mechanistic basis for the observations reported here. We hope the reviewer will recognize that the present manuscript represents a substantial body of work, and that it is important to publish these foundational data so that we and others can build on the studies reported here.

Minor Concerns:

1. Structural Analysis of TUG: The structural analysis of TUG is well-conducted. However, I recommend including some level of quantification and incorporating reference markers for secretory compartments to strengthen the results.

Similar to the full-length TUG protein, we show that of IDR1-mCherry shows colocalization with Em-ERGIC53 and GFP-Stx12 (Fig. 4f, g) and is distinct from the ERES marker Sec23-GFP (Supplementary Fig. 6c) and excluded from the Golgi marked using GMAP210-mNG (Supplementary Fig. 6d).

2. Phase-Boundary Analysis: Incorporating phase-boundary analysis would enhance the data presented in Figures 4i and 4j.

The images we have shown (now in Figs. 5i,j of the revised manuscript) were indeed a part of phase-boundary analysis. At protein concentrations higher than what we have shown, or at higher Ficoll concentrations, we continue to see subtle differences in the degree of droplet formation between mCherry-TUG- Δ 1-183 and mCherry-TUG. We think that, rather than showing a simple yes or no phase diagram, the images shown here make a strong case to highlight the differences between the full-length and the N-terminal deletion mutant. We did not observe condensates at lower protein concentrations, either for mCherry-TUG- Δ 1-183 or for the intact protein.

Responses to Reviewer Comments

We appreciate the three reviewers' thoughtful analysis and comments. All comments are reproduced below with our point-by-point responses in blue underneath each comment.

Reviewer #1 (Remarks to the Author):

The authors have responded to all points that I initially raised. Although it is a pity that some of the experiments I asked for could not be done (e.g. the RUSH assay or the trafficking of an endogenous cargo). I nevertheless thank the authors for trying to solve these questions. Overall, the manuscript improved and the critical points have been addressed to my satisfaction. I therefore have no further points and think this work should be published now.

We thank the reviewer for this positive assessment and for supporting publication of the revised manuscript.

Reviewer #2 (Remarks to the Author):

In the resubmitted version of their manuscript, the authors have addressed the concerns of this reviewer. The attention to detail in the text and responses is appreciated. The additional data the authors provided comparing mCherry vs. mNeonGreen is convincing and shows that the fluorophore has negligible effect on the phase separation of TUG. The FRAP experiments also provide additional evidence that TUG is forming condensates, as the fluorescence recovery is consistent with condensed material on intracellular membranes. The additional graphic models in the supplementary material are also helpful for visualizing the pathway. Finally, the method for calculating distances in supplemental figure 4 is an excellent addition and clearly explains the methodology while providing a road map for other researchers to apply this method to their research. With the additional work on this manuscript, this reviewer now supports publication in Nature Communications.

We thank the reviewer for this positive assessment and for supporting publication of the revised manuscript.

Reviewer #3 (Remarks to the Author):

Overall, the revised manuscript has significantly improved. The authors have addressed most of my concerns properly. The only concern remains related to the secretory flux experiment. The authors have invested a significant amount of time and efforts to develop assays for a new RUSH reporter in MEFs. I understand that sometimes the RUSH system simply does not work in certain cells, as we have encountered similar situation in our own work. That said, there are many other existing RUSH reporters to serve this purpose. Additionally, the authors also have a HeLa cell system for addressing the secretory flux upon TUG depletion. These experiments are crucial to substantiate the claim that TUG KO MEFs have greater ER-Golgi cargo flux (line 168-171).

In its current form, the data seem to point toward a cargo-specific effect: collagen and CFTR secretion were significantly reduced, while small and soluble cargo proteins appeared to be unaffected (Figure 6). If TUG truly acts as a general "brake" on early secretory pathway

trafficking, one would expect a broader impact across cargo types. Therefore, the findings are more consistent with a selective role of TUG in regulating certain cargoes, rather than a general inhibition of ER–Golgi transport.

We thank the reviewer for this overall positive assessment of the revised manuscript. We agree that there are cargo-specific effects of TUG deletion, and that effects on collagen and CFTR secretion are much more marked than effects on small, soluble cargo proteins. Accordingly, we have emphasized this point more clearly in the revised text.

Although we can study effects of TUG deletion in HeLa cells, effects in fibroblasts (MEFs) are much more robust. We note that TUG abundance is ~30-fold greater in MEFs than in HeLa cells, as shown in Supplementary Fig. 2A. This may explain why there is a marked effect of TUG deletion on Golgi morphology in MEFs, whereas our previous work using HeLa cells detected only subtle alterations. We also note that accelerated ER-Golgi flux of a soluble secretory reporter, PAUF, was increased by TUG deletion in MEFs (Fig. 2), but that the overall secretion of small, soluble cargoes was unchanged (Fig. 6). This may reflect a rate-limiting step in secretion of small, soluble cargoes that is subsequent to the ERGIC. We have now added these points to the discussion. We think the overall manuscript is strengthened by the clearer description included in this revised version, and we hope that its publication will stimulate future studies to more fully define the mechanism for cargo-selective effects.